# Amyloid-like aggregating proteins cause lysosomal defects in neurons via gain-of-function toxicity

Irene Riera-Tur[1,2,*] , Tillman Schäfer[3,*] , Daniel Hornburg[4,5,*] , Archana Mishra[1], Miguel da Silva Padilha[1,2], Lorena Fernández-Mosquera[6], Dennis Feigenbutz[1,2], Patrick Auer[1,2], Matthias Mann[4] , Wolfgang Baumeister[3], Rüdiger Klein[1] , Felix Meissner[5,10], Nuno Raimundo[7] , Rubén Fernández-Busnadiego[3,8,9,†] , Irina Dudanova[1,2,†]

The autophagy-lysosomal pathway is impaired in many neurodegenerative diseases characterized by protein aggregation, but the link between aggregation and lysosomal dysfunction remains poorly understood. Here, we combine cryo-electron tomography, proteomics, and cell biology studies to investigate the effects of protein aggregates in primary neurons. We use artificial amyloid-like $\beta$-sheet proteins ($\beta$ proteins) to focus on the gain-of-function aspect of aggregation. These proteins form fibrillar aggregates and cause neurotoxicity. We show that late stages of autophagy are impaired by the aggregates, resulting in lysosomal alterations reminiscent of lysosomal storage disorders. Mechanistically, $\beta$ proteins interact with and sequester AP-3 $\mu$1, a subunit of the AP-3 adaptor complex involved in protein trafficking to lysosomal organelles. This leads to destabilization of the AP-3 complex, missorting of AP-3 cargo, and lysosomal defects. Restoring AP-3$\mu$1 expression ameliorates neurotoxicity caused by $\beta$ proteins. Altogether, our results highlight the link between protein aggregation, lysosomal impairments, and neurotoxicity.

## Introduction

The autophagy-lysosomal system is a major cellular degradation pathway for long-lived proteins, macromolecular complexes, and damaged organelles (Settembre et al, 2013; Finkbeiner, 2020). Defects in this system cause a number of severe disorders known as lysosomal storage diseases, often characterized by early-onset neurodegeneration (Fraldi et al, 2016; Platt et al, 2018). In addition,

lysosomal function is compromised in many age-related neurodegenerative disorders, such as amyotrophic lateral sclerosis, Alzheimer's, Parkinson's, and Huntington's disease (Abeliovich & Gitler, 2016; Fraldi et al, 2016; Taylor et al, 2016; Wang et al, 2018; Finkbeiner, 2020; Nixon, 2020). Another convergent feature of late-onset neurodegenerative diseases is protein misfolding and aggregation, which leads to accumulation of toxic protein species and neuronal demise (Soto & Pritzkow, 2018). As autophagy represents an important pathway for removal of aggregated proteins, dysfunction of the autophagy-lysosomal system facilitates the build-up of aggregates (Wang et al, 2018; Finkbeiner, 2020). Conversely, aggregating proteins themselves might directly interfere with the normal function of the autophagy-lysosomal machinery (Cuervo et al, 2004; Winslow et al, 2010; Wong & Holzbaur, 2014). However, until now the mechanistic link between aggregation and lysosomal impairments is not completely understood.

One difficulty in studying the role of protein aggregation in cellular dysfunction is the overlap of gain- and loss-of-function effects resulting from protein misfolding. On one hand, reduction in the cellular pool of the correctly folded form of an aggregating protein results in a partial loss of its native function. At the same time, the misfolded conformation of the protein can engage in aberrant interactions with cellular membranes and with other proteins, causing gain-of-function toxicity (Olzscha et al, 2011; Winner et al, 2011; Kim et al, 2016; Yang & Hu, 2016; Bauerlein et al, 2017; Chiti & Dobson, 2017). These loss- and gain-of-function effects occur in parallel and are difficult to disentangle, especially because the native functions of many aggregating proteins are still poorly understood (Winklhofer et al, 2008; Saudou & Humbert, 2016;

[1]Department of Molecules–Signaling–Development, Max Planck Institute of Neurobiology, Martinsried, Germany   [2]Molecular Neurodegeneration Group, Max Planck Institute of Neurobiology, Martinsried, Germany   [3]Department of Molecular Structural Biology, Max Planck Institute of Biochemistry, Martinsried, Germany   [4]Department of Proteomics and Signal Transduction, Max Planck Institute of Biochemistry, Martinsried, Germany   [5]Experimental Systems Immunology Group, Max Planck Institute of Biochemistry, Martinsried, Germany   [6]The William Harvey Research Institute, Barts and The London School of Medicine and Dentistry, Queen Mary University of London, London, UK   [7]Department of Cellular and Molecular Physiology, Penn State College of Medicine, Hershey, PA, USA   [8]Institute of Neuropathology, University Medical Center Goettingen, Goettingen, Germany   [9]Cluster of Excellence "Multiscale Bioimaging: from Molecular Machines to Networks of Excitable Cells" (MBExC), University of Goettingen, Goettingen, Germany   [10]Department of Systems Immunology and Proteomics, Institute of Innate Immunity, Medical Faculty, University of Bonn, Bonn, Germany

Correspondence: idudanova@neuro.mpg.de; ruben.fernandezbusnadiego@med.uni-goettingen.de
Tillman Schäfer's present address is Cryo-EM Facility, Max Planck Institute of Biochemistry, Martinsried, Germany.
Daniel Hornburg's present address is Department of Genetics, Stanford University School of Medicine, Stanford, CA, USA.
*Irene Riera-Tur, Tillman Schäfer, and Daniel Hornburg contributed equally to this work.
†Rubén Fernández-Busnadiego and Irina Dudanova contributed equally to this work.

Brothers et al, 2018). To overcome this challenge, here we take advantage of artificial proteins (hereafter β proteins), which have been rationally designed to form antiparallel β-sheets because of an alternating pattern of polar and non-polar amino acid residues (West et al, 1999). Antiparallel β-sheet structure is an important general property of natural aggregating proteins, as demonstrated for Aβ, polyQ proteins, pathological tau, and α-synuclein (Chiti & Dobson, 2017; Hartl, 2017). β proteins spontaneously assemble into amyloid-like fibrils in vitro (West et al, 1999). When expressed in cells, β proteins form inclusions and compromise cellular viability (Olzscha et al, 2011; Woerner et al, 2016; Vincenz-Donnelly et al, 2018; Frottin et al, 2019). Importantly, although being structurally similar to natural amyloids, β proteins do not possess any biological function, and therefore provide an excellent tool to investigate toxic gain-of-function effects of aggregation in the absence of any loss-of-function phenomena.

In this study, we use a combination of cryo-electron tomography (cryo-ET), proteomics, and cell-biological approaches to study the impact of β proteins on primary neurons. We find that β protein aggregation impairs autophagy, leads to accumulation of enlarged lysosomes with undigested cargo, and induces neurotoxicity. Our data suggest that these defects are mediated, at least in part, by the sequestration of a subunit of the AP-3 adaptor complex within the aggregates. Taken together, our findings link lysosomal impairments to toxic gain-of-function of protein aggregates.

# Results

## β protein aggregation causes toxicity in primary neurons

To investigate the effects of β protein aggregation in neurons, we used two β proteins, β4 and β23, which differ in primary amino acid sequence, but adopt similar β-sheet structures (West et al, 1999). The proteins were tagged with a myc epitope and fused to mCherry (β4-mCherry and β23-mCherry). We expressed β proteins in dissociated murine cortical cultures using transfection or lentiviral transduction (see Table S1 for a summary of experimental conditions). Upon transfection of β4-mCherry and β23-mCherry, we observed formation of abundant cytoplasmic aggregates of irregular shape (Fig 1A and B). To assess whether the presence of β proteins caused toxicity, we stained transfected cultures for the apoptotic marker cleaved caspase-3 (Fig 1C). Both β4-mCherry and β23-mCherry induced a significant increase in the number of cleaved caspase-3–positive cells at DIV 10+3 (Fig 1D). We observed similar results with lentivirally transduced neurons, where both β4-mCherry and β23-mCherry aggregated and accumulated in the insoluble fraction from DIV 10+4 (Fig S1A–D). Significant toxicity was observed for both β proteins from DIV 10+6 onwards, corresponding to appearance of abundant aggregates (Fig S1E).

We further assessed dendritic complexity, as it can be affected by pathological aggregates (May et al, 2014). These experiments were performed on transfected hippocampal neurons because of their uniform morphology. Sholl analysis revealed a significant reduction in dendritic complexity in neurons transfected with either β4-mCherry or β23-mCherry (Fig 1E and F). Altogether, these data indicate that β proteins aggregate and cause toxicity in primary neurons.

## Ultrastructure of neuronal β protein aggregates

We used cryo-ET to explore the ultrastructure of β protein aggregates and their potentially toxic cellular interactions in neurons. This technique allows investigating the structure of protein aggregates and their impact on the cellular milieu in close-to-native conditions at molecular resolution (Bauerlein et al, 2017; Wagner et al, 2017; Gruber et al, 2018; Guo et al, 2018; Trinkaus et al, 2021). Primary cortical neurons were grown on electron microscopy (EM) grids, transfected with β proteins, and vitrified by plunge-freezing. Cryo-correlative microscopy allowed targeting the aggregates for cryo-focused ion beam (cryo-FIB) milling (Rigort et al, 2012) and subsequent cryo-ET using a Volta phase plate (Danev et al, 2014) (Fig S2A–F).

Analysis of cryo-electron tomograms revealed that all β protein aggregates displayed fibrillar morphology (for numbers of analyzed aggregates, see Table S2) and consisted of an apparently disordered network of very thin fibrils (Fig 2A–D). The fibril diameters were similar for both β proteins (β4-mCherry, 4.1 ± 1.6 nm; β23-mCherry, 3.6 ± 1.1 nm; n = 30 fibrils in both cases; unpaired $t$ test, n. s.) (Fig 2E). The fibrils were highly curved and branched, similar to those observed in vitro (Olzscha et al, 2011). The fibrillar network encapsulated additional electron-dense structures that may correspond to cellular proteins sequestered by the aggregates (Olzscha et al, 2011; Woerner et al, 2016), such as ribosomes (Fig 2B). No substantial differences were observed between neuronal aggregates of β4-mCherry and β23-mCherry, or between aggregates found in neurons and in HeLa cells (Figs 2 and S3 and Table S2).

β protein aggregates were often found in direct contact with cellular membranes, especially those of the ER (Figs 2, S3, and S4F). In some cases, ER tubes surrounded the aggregate periphery and tunnelled through its interior (Fig S4F), similar to previous observations for polyQ, α-Synuclein and heat-shock induced aggregates (Bauerlein et al, 2017; Wagner et al, 2017; Gruber et al, 2018; Trinkaus et al, 2021). However, in contrast to polyQ fibrils (Bauerlein et al, 2017), β protein fibrils did not appear to deform cellular membranes (Figs 2C and S3A). The ER around the aggregates often engaged in membrane contact sites with mitochondria (Figs 2D and S3B), as observed for stress-induced and polyQ aggregates (Zhou et al, 2014; Gruber et al, 2018). Thus, despite the distinct characteristics of aggregates formed by different proteins, β protein aggregates reproduced some features of natural amyloids.

## β protein expression leads to defects of lysosomal morphology

Besides the presence of aggregates, the most striking ultrastructural feature of β protein–expressing neurons was the accumulation of large endo-lysosomal organelles (Figs 3 and S4D–I). In control cells, a variety of endo-lysosomes was observed, including tubular early endosomes, multivesicular bodies, and autolysosomes containing membranous cargo (Figs 3A and B and S4A–C). Although all those species were also found in β protein–expressing cells, large (>1 μm in diameter) cargo-loaded autolysosomes were enriched. No β protein aggregates were observed within these organelles in tomograms or lamella overview images (for numbers of analyzed lysosomes, see Table S2). Instead, autolysosomes often contained extensive stacks of parallel membranes and smaller

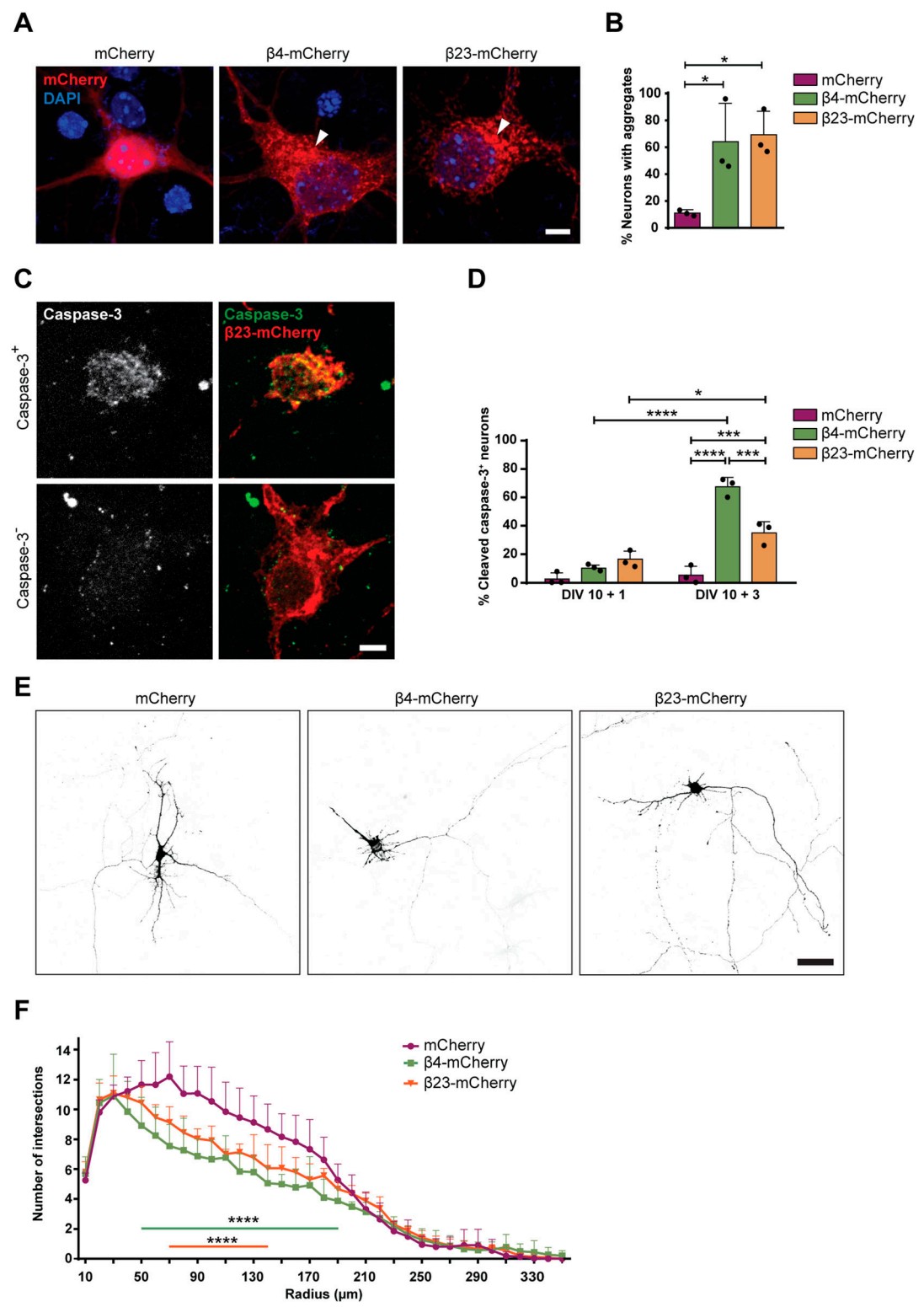

**Figure 1. β proteins aggregate and cause toxicity in transfected primary neurons.**
**(A)** Transfected cortical neurons at DIV 10+1. Arrowheads point to β protein aggregates. **(B)** Percentage of transfected neurons bearing aggregates at DIV 10+1 (n = 3 independent experiments, 25–45 cells/condition/experiment; one-way ANOVA with Dunnett's post hoc test). **(C)** Examples of DIV 10+1 β23-mCherry neurons positive (top) and negative (bottom) for cleaved caspase-3. **(D)** Percentage of transfected neurons positive for cleaved caspase-3 (n = 3 independent experiments, 25–45 cells/ condition/experiment; two-way ANOVA with Tukey's post hoc test). **(E)** Examples of DIV 10+2 primary hippocampal neurons transfected with mCherry (left), β4-mCherry (middle), or β23-mCherry (right). Images are colour-inverted with mCherry fluorescence shown in black. Note that the β protein cells have fewer primary dendrites. **(F)** Sholl analysis reveals reduced dendritic complexity in the presence of β proteins (n = 3 independent experiments, 10–30 cells/condition/experiment; two-way ANOVA with Tukey's post hoc test). Scale bars, 5 μm in (A and C); 50 μm in (E). Data information: Data are presented as mean ± SD. *P < 0.05; ***P < 0.001; ****P < 0.0001.

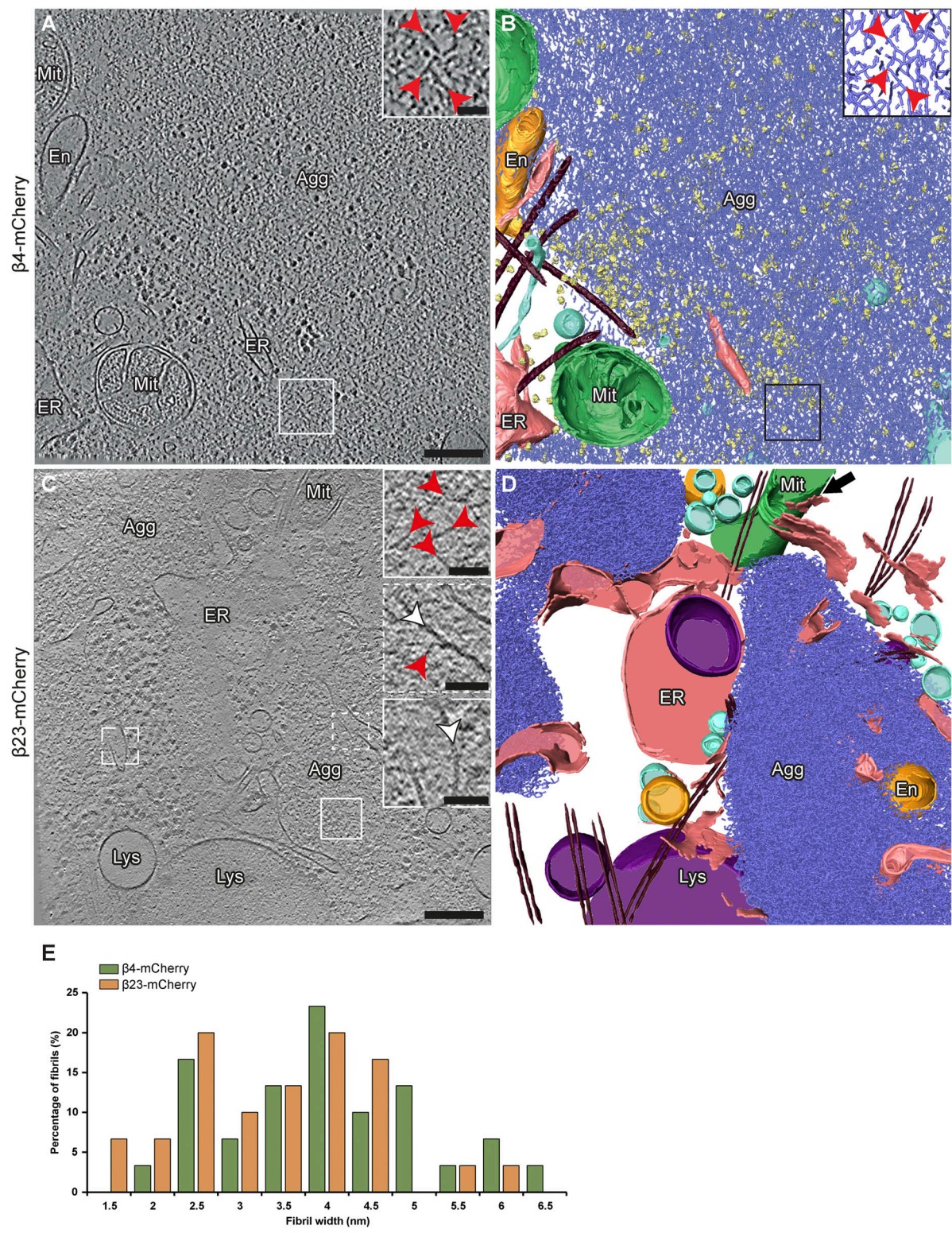

**Figure 2. Ultrastructure of β protein aggregates in primary neurons.**
**(A)** Tomographic slice of a β4-mCherry aggregate in transfected DIV 6+1 cortical neurons. **(B)** 3D rendering of the tomogram shown in (A). **(C)** Tomographic slice of two β23-mCherry aggregates in neurons. **(D)** 3D rendering of the tomogram shown in (C). The areas marked by the boxes are magnified in the insets. Red arrowheads point to β protein fibrils. White arrowheads point to intracellular membranes. Note that intracellular membranes in contact with β protein fibrils (middle inset in C) are not deformed and do not differ from membranes that are not in direct contact with aggregates (lower inset in C). Agg, β protein aggregate; En, endosome; ER, endoplasmic reticulum; Lys, lysosome; Mit, mitochondrion. Black arrow in (D) indicates ER - mitochondria contact site. β protein fibrils, blue; mitochondria, green; ER membranes, salmon; endosome, gold; vesicles, cyan; ribosomes, yellow; microtubules, brown. **(E)** Histogram of β4-mCherry and β23-mCherry fibril diameters (n = 30 β4-mCherry fibrils and 30 β23-mCherry fibrils, from three tomograms each). Scale bars in (A, C), 200 and 50 nm (insets).

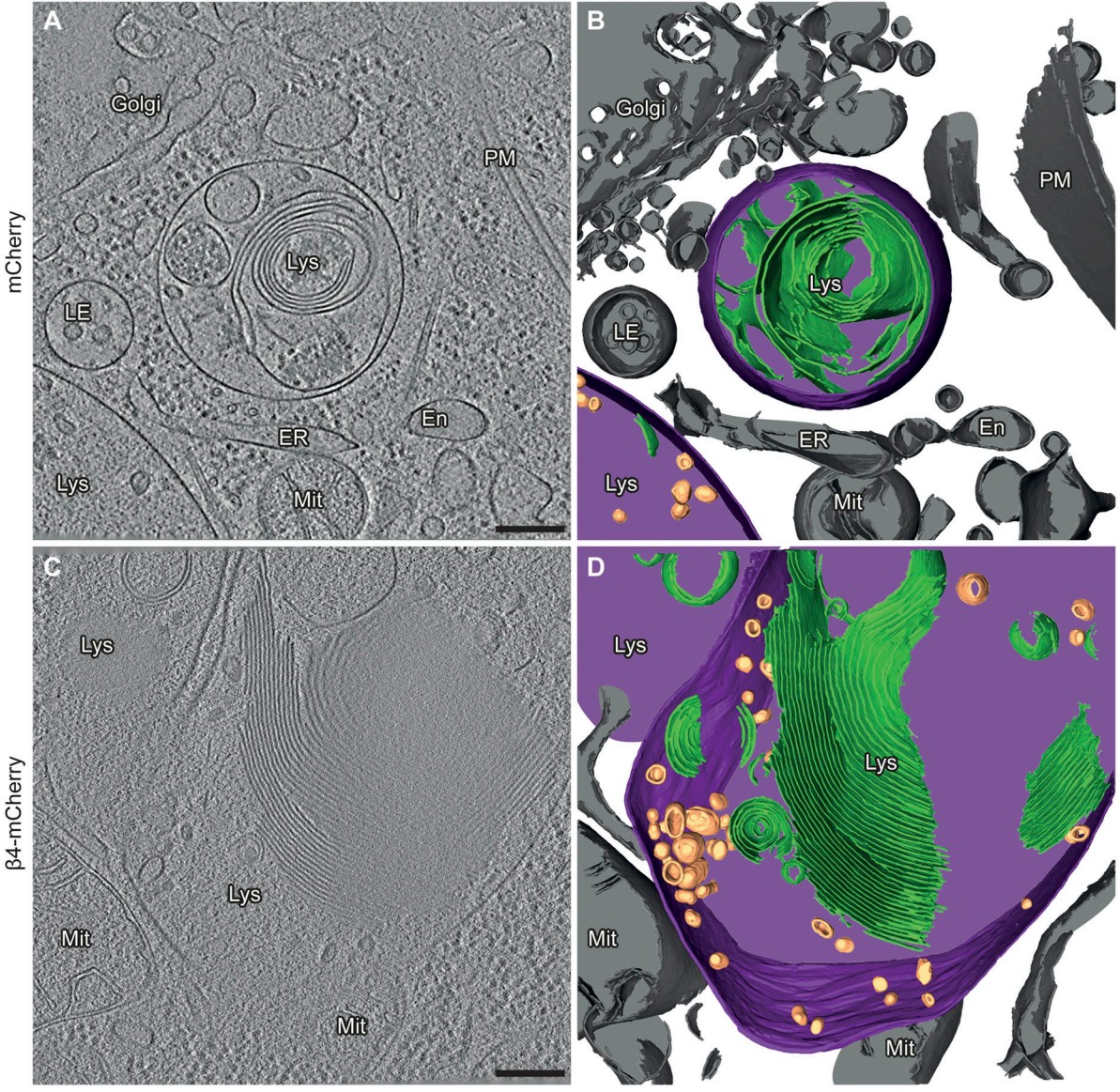

**Figure 3. Aberrant lysosomal ultrastructure in β protein–expressing neurons.**
**(A)** Example of a lysosome in a tomogram from a DIV 6+1 cortical neuron transfected with mCherry. **(B)** 3D rendering of the tomogram shown in (A). **(C)** Example of a lysosome in a neuron transfected with β4-mCherry. Note the presence of abundant membrane stacks and electron-dense material within the lysosome. **(D)** 3D rendering of the tomogram shown (C). En, endosome; ER, endoplasmic reticulum; LE, late endosome; Lys, lysosome; Mit, mitochondrion; PM, plasma membrane. Lysosomal membrane, purple; membrane stacks within the lysosomes, green; intraluminal vesicles, gold; other cellular membranes, grey. For additional examples see Fig S4. Scale bars in (A, C), 200 nm.

vesicles, together with an electron-dense lumen suggestive of a high protein concentration (Figs 3C and D and S4D and H). Interestingly, similar structures were reported in Alzheimer's disease (Nixon et al, 2005; Gowrishankar et al, 2015), in Parkinson's disease, and other synucleinopathies (Crews et al, 2010; Dehay et al, 2012; Usenovic et al, 2012) and in conditions of impaired lysosomal degradation (Abraham et al, 1968; Spaet et al, 1983; Fernandez-Mosquera et al, 2019).

Consistently with cryo-ET, light microscopy experiments with LysoTracker-loaded neurons showed that β protein expression led to an increase in lysosomal size (Fig 4A and B). Compared with mCherry cells, lysosomes with a diameter larger than 1 μm were

~3.5-fold more abundant in both β4-mCherry and β23-mCherry cells. However, the total number of LysoTracker-positive puncta per cell was reduced (Fig 4C). In agreement with cryo-ET findings, no significant colocalization was observed between β protein aggregates and LysoTracker-positive organelles (Fig 4A and D). In summary, our cryo-ET and light microscopy data show that β protein aggregation leads to the accumulation of enlarged, cargo-rich autolysosomes, although the aggregates themselves do not appear to build up inside these organelles. This is compatible with a scenario in which autophagic cargo is successfully delivered to lysosomes, but lysosomal degradation is defective.

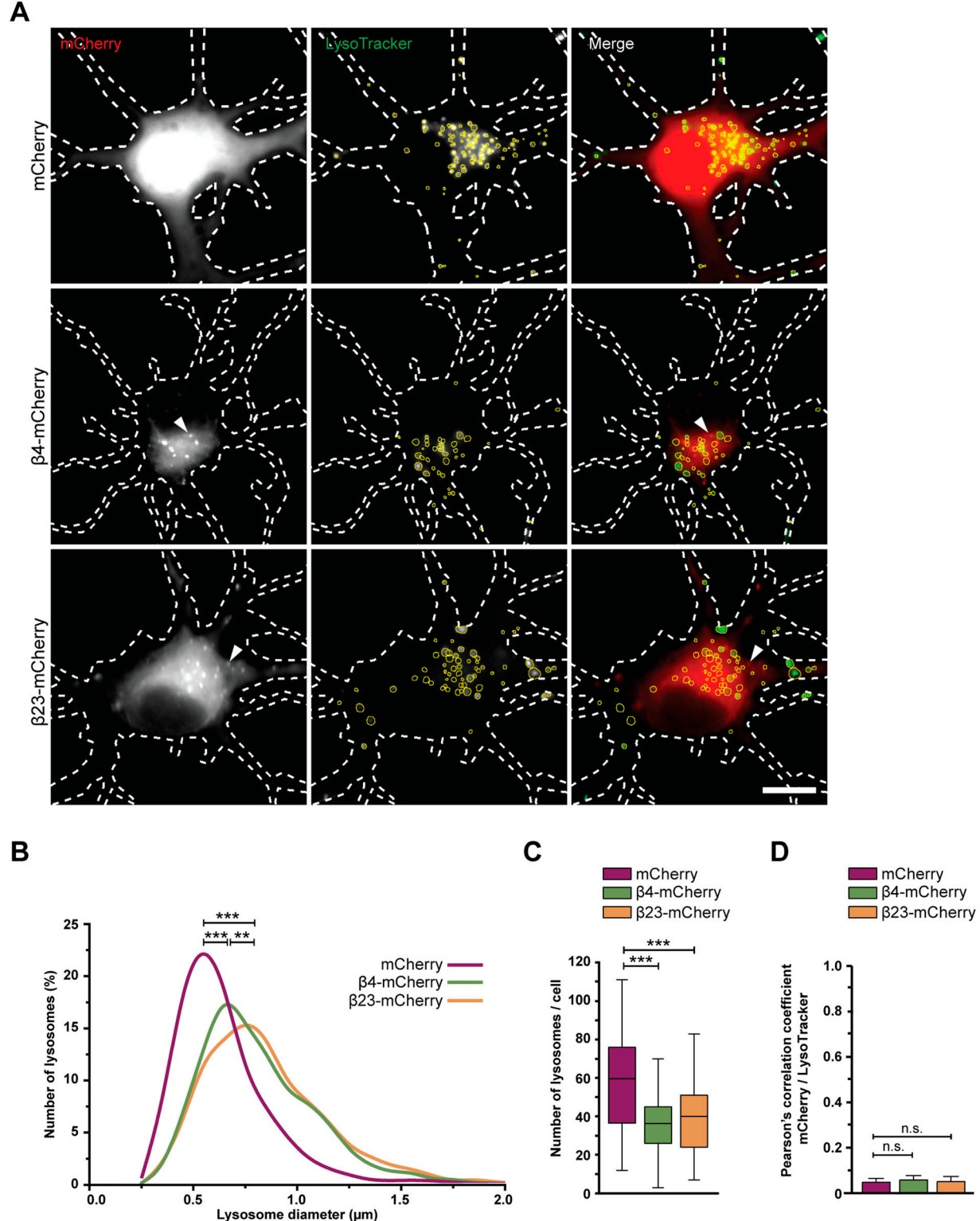

**Figure 4. Defects of lysosomal morphology in the presence of β proteins.**
**(A)** Fluorescence images of DIV 6+1 primary cortical neurons transfected with mCherry (top), β4-mCherry (middle), or β23-mCherry (bottom) and incubated with LysoTracker Green. White dashed lines show the contours of the neurons. Arrowheads point to β protein aggregates. Yellow circles outline lysosomes. **(B)** Distribution of the lysosomal size in control and β protein–expressing neurons (mCherry, n = 2,148 lysosomes from 36 cells; β4-mCherry, n = 1,595 lysosomes from 44 cells; β23-mCherry, n = 1,838 lysosomes from 46 cells; from four independent experiments; two-tailed Mann–Whitney test). **(C)** Box plot showing the number of lysosomes per neuron (mCherry, n = 36 cells; β4-mCherry, n = 44 cells; β23-mCherry, n = 46 cells; from four independent experiments; two-tailed Mann–Whitney test). **(D)** Quantification of Person's correlation coefficient between the mCherry and LysoTracker signal (n = 36 mCherry cells, 39 β4-mCherry cells, 36 β23-mCherry cells; one-way ANOVA with Tukey's post hoc test). Scale bar in (A), 10 μm. Data information: Data in (C) are presented as box plots with whiskers indicating minimal and maximal values. Data in (D) are presented as mean ± SD. **$P < 0.01$; ***$P < 0.001$; n.s., not significant.

## β proteins impair the autophagy-lysosomal pathway

To assess possible dysfunctions of the autophagy-lysosomal pathway, we estimated autophagic flux by monitoring LC3-II, an autophagy-related protein that is itself degraded by lysosomes. First, we quantified LC3-II turnover in β23-mCherry–transfected HeLa cells (Fig S5A) in the presence and absence of 50 μM chloroquine, which blocks lysosomal degradation (Poole & Ohkuma, 1981; Mauthe et al, 2018). In control mCherry cells, application of chloroquine resulted in a significant, twofold increase in LC3-II, whereas no significant accumulation of LC3-II occurred in β23-mCherry cells (Fig 5A and B). Accordingly, the ratio of LC3-II levels with and without chloroquine was significantly reduced in β23-mCherry cells (Fig 5C), indicative of a partial block in the autophagy-lysosomal pathway.

Second, we assessed autophagic flux in primary neurons with the help of a tandem mCherry-GFP-LC3 reporter (Leeman et al, 2018). In the low pH conditions of functional lysosomes, the GFP fluorescence of the reporter is quenched and the organelles appear red. In contrast, less acidic organelles are marked with both GFP and mCherry and appear yellow (Fig 5D). For these experiments, we used myc-tagged β4 protein not fused to mCherry (from here on myc-β4). A rationally designed α-helical protein with similar amino acid composition (myc-α-S824) (Wei et al, 2003; Olzscha et al, 2011) served as a control. Expression of myc-β4, but not myc-α-S824, resulted in aggregation and caused toxicity in neurons (Fig S5B and C). When each of these proteins was co-transduced into primary neurons together with the mCherry-GFP-LC3 reporter, the total number of fluorescent puncta was not different between myc-β4 and myc-α-S824 neurons (Fig 5E and F). However, in myc-β4 cells, the number of yellow (non-acidic) puncta was increased, whereas the number of red-only (acidic) puncta was decreased (Fig 5G). These results argue against an overall increase in autophagy induction in the presence of the aggregating protein and confirm a defect in lysosomal degradation.

To further test whether autophagy induction was altered, we monitored the levels of the early autophagy markers Beclin 1 and ATG5 (Yamamoto & Yue, 2014) in HeLa cells transfected with β23-mCherry. These markers were not significantly different in β23-mCherry and mCherry control cells (Fig 5H–I), suggesting that autophagosome formation was not affected by β proteins. Consistently, early autophagosomes were rarely observed by cryo-ET in either control or β protein–expressing neurons (Fig S4E, numbers of observations are provided in Table S2). In contrast, real-time PCR revealed that β protein expression induced an up-regulation of transcripts of numerous lysosomal proteins (Fig 5J), a characteristic signature of aberrant lysosomal storage (Sardiello et al, 2009; Settembre et al, 2011). Taken together, our data suggest that β protein aggregates do not affect early stages of autophagy, but impair lysosomal function.

## Interactome of β proteins in neurons

To search for the molecular causes of the observed lysosomal alterations, we characterized the interactome of β proteins in neurons using quantitative label-free mass spectrometry (MS). In these experiments, we also used another mCherry-fused artificial amyloid-like protein called β17. The aggregation propensity of the proteins increases from β4 to β17 to β23 (West et al, 1999; Olzscha et al, 2011). Primary cortical neurons lentivirally transduced with β proteins or mCherry were harvested at DIV 10+4, a time point when β proteins are still largely soluble and do not cause massive cell death (Fig S1). Immunoprecipitation against mCherry was then performed to isolate interactors, and the total proteome was also analyzed.

Using highly stringent criteria, we identified 30 β4-mCherry interactors, 54 β17-mCherry interactors and 59 β23-mCherry interactors, with an extensive overlap between the three β proteins (Fig 6A–E and Table S3). In addition to confirming interactions found previously in non-neuronal cells (Olzscha et al, 2011), our approach uncovered a number of neuron-specific interactors, including several proteins linked to neurodegenerative disorders, such as Aimp1 (Armstrong et al, 2014), Mark4 (Rovelet-Lecrux et al, 2015), and Ppp2r5d (Louis et al, 2011) (Table S3). The most highly enriched protein in the interactome of all three β proteins was AP-3μ1, the medium subunit of the heterotetrameric AP-3 adaptor complex (Fig 6B–E). The AP-3 complex is involved in intracellular trafficking of transmembrane proteins, including protein transport to lysosomes (Newell-Litwa et al, 2007), and the medium subunit is responsible for cargo recognition (Ohno et al, 1998). Other prominent common interactors were Ccdc88a and Crmp1, two proteins involved in neuronal development and synaptic plasticity (Enomoto et al, 2009; Yamashita & Goshima, 2012; Nakai et al, 2014). Of note, Ccdc88a plays a role in intracellular trafficking, among other functions (Le-Niculescu et al, 2005). Crmp1 is also found within mutant Huntingtin inclusions, and suppresses neurotoxicity in Huntington's disease models (Stroedicke et al, 2015).

Annotation enrichment analysis of the interactors revealed their involvement in key signaling pathways, such as "protein serine/threonine kinase activity," "GTPase regulator activity," and "transport" (Fig S6A–C). Another enriched category was "microtubule," consistent with the impaired morphology of β protein–expressing neurons (Fig 1E and F). In addition to these general cellular pathways, we found neuron-specific categories, such as "synapse," "postsynaptic cell membrane," and "neuron projection" (Fig S6A–C). In line with previous proteomic investigations of other aggregates (Olzscha et al, 2011; Kim et al, 2016; Woerner et al, 2016; Hosp et al, 2017), we observed a gradual increase in low-complexity regions in β protein interactomes that paralleled their aggregation propensity (Fig S6D).

Analysis of the total proteome of transduced neurons (which includes the proteins interacting with the aggregates) did not reveal significant changes in the levels of any of the β protein interactors (Fig S7A–D and Tables S3 and S4). This indicates that their presence in the interactome was not merely a result of their increased amounts in the cells, nor was their sequestration markedly compensated by increased expression. The unaltered total levels of the interactors, together with their enrichment in the interactome, suggest that the biologically active cellular pool of these proteins might be reduced, potentially leading to functional impairments.

## AP-3μ1 is sequestered by β protein aggregates, compromising the integrity of the AP-3 complex

For further studies, we focused on AP-3μ1 as the most highly enriched β protein interactor. This protein made up 26% of the β4-mCherry–interacting complexes, 17% for β17-mCherry, and 10% for

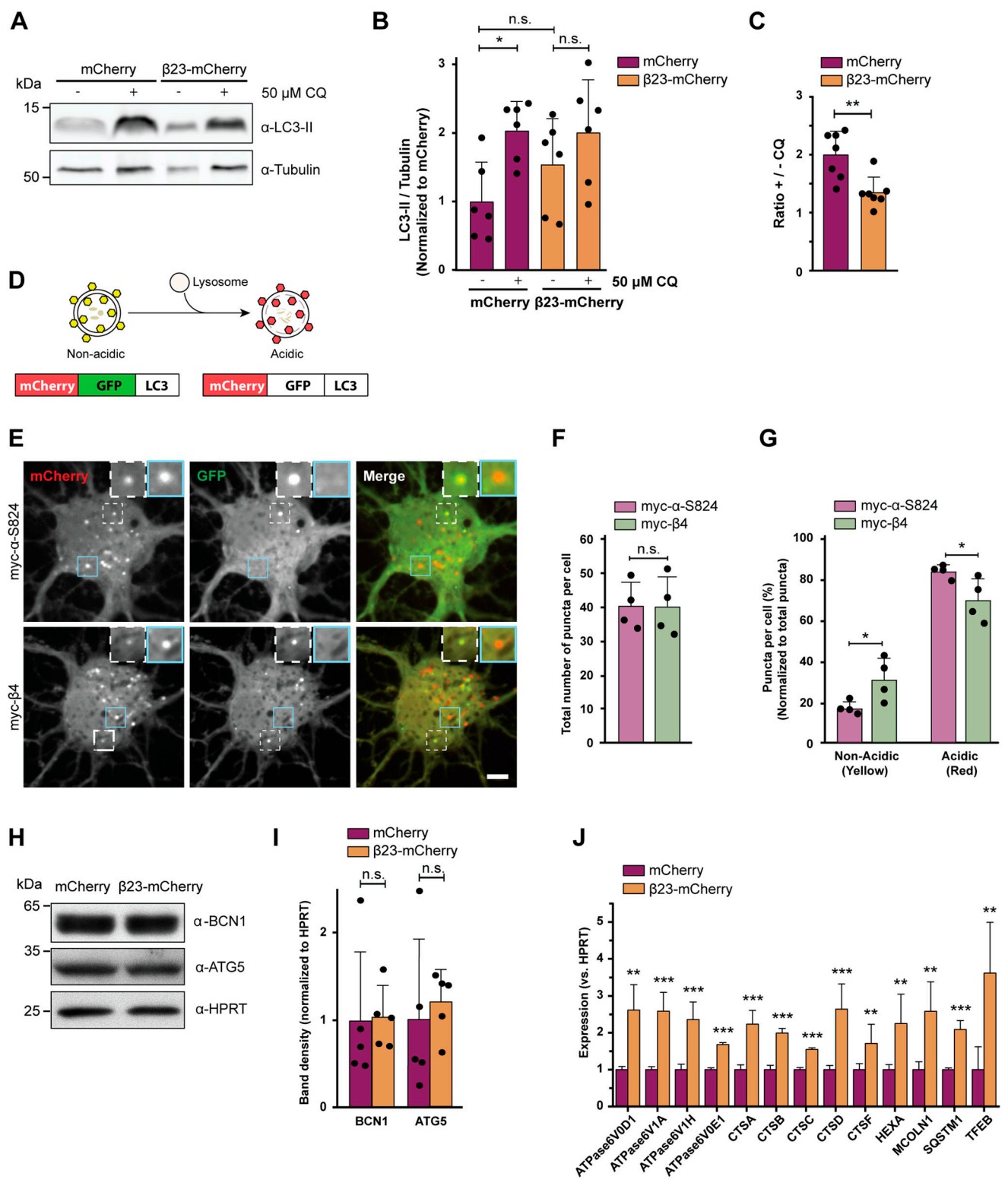

**Figure 5.  β proteins impair autophagy.**
**(A)** Western blot for LC3B-II in HeLa cell lysates under control conditions and upon treatment with 50 μM chloroquine. Tubulin was used as a loading control.
**(B)** Quantification of LC3B-II levels. All conditions were normalized to mCherry (n = 6 independent experiments; one-way ANOVA with Tukey's multiple-comparisons test).
**(C)** Quantification of the ratio of LC3B-II levels in cells treated or not treated with chloroquine. Same data were analyzed as shown in (B) (n = 6 independent experiments; two-tailed *t* test with Welch's correction). **(D)** mCherry-GFP-LC3 reporter appears yellow in non-acidic and red in acidic organelles due to quenching of GFP fluorescence at low pH. **(E)** Single plane images of the reporter signal in DIV 10+3 neuronal cultures transfected with myc-α-S824 (top) and myc-β4 (bottom). Examples of non-acidic and

β23-mCherry (Fig 6F–H), whereas its amount in the total proteome was not significantly changed (Fig S7B–D). In agreement with our MS data, both β4-mCherry and β23-mCherry co-immunoprecipitated with endogenous AP-3μ1 in neurons (Fig S7E). As the interactome experiments were performed with the soluble fraction of neuronal lysates, we asked whether AP-3μ1 is also present within β protein aggregates. Using immunofluorescence, we found a high degree of colocalization between co-transfected AP-3μ1 and β protein aggregates (Fig 7A and B). Cell fractionation experiments further revealed a significant accumulation of AP-3μ1 in the pellet fraction of β protein neurons (Fig 7C–E). These data show that AP-3μ1 not only interacts with soluble β proteins, but is sequestered by β protein aggregates.

To test whether AP-3μ1 is also sequestered by natural amyloids, we analyzed its colocalization with aggregates formed by the Tau repeat domain (TauRD, residues 244–372 with the point mutations P301L/V337M, fused to EYFP), which mediates the assembly of tau fibrils (Crowther et al, 1992). Neurons transduced with TauRD-EYFP were treated with recombinant TauRD seeds to induce aggregation (Fig 7F) (Yuste-Checa et al, 2021). We observed significantly higher colocalization between TauRD-EYFP and endogenous AP-3μ1 in neurons treated with seeds compared with non-seeded control cultures (Fig 7F and G). In addition, AP-3μ1 significantly accumulated in the pellet fraction of the seeded cultures (Fig 7H–J). Taken together, these data suggest that AP-3μ1 can be substantially sequestered by different types of aggregates. As AP-3μ1 is not highly abundant in neurons (Fig S7F), its sequestration might result in a partial loss of function.

Defects in single subunits of the AP-3 complex have been previously shown to result in destabilization of the entire complex and premature degradation of its other subunits (Kantheti et al, 1998). We therefore quantified the levels of two other AP-3 complex subunits in β protein neurons: AP-3δ1, one of the large subunits, and AP-3σ1, the smll subunit. Although these two subunits did not directly engage with β proteins (Table S3), their levels significantly decreased (Fig 8A and B) in β protein cells (Fig S1A–D). These results suggest that the partial loss of the medium subunit leads to lower amounts of other AP-3 subunits and likely to a reduction in the levels of intact AP-3 complex in the cell. AP-3 complex deficiency is known to cause defects in biogenesis and function of lysosome-related organelles in mice and in humans, for example, in the rare genetic disorder Hermansky–Pudlak syndrome (Dell'Angelica & Bonifacino, 2019). Thus, we hypothesized that AP-3 loss of function may play a role in the lysosomal defects and neurotoxicity caused by β proteins.

### AP-3 function is impaired by β proteins, leading to lysosomal defects

To assess the function of the AP-3 complex in the presence of β proteins, we determined the localization of the lysosomal protein Lamp1, a known cargo of AP-3. Lamp1 is at least partially trafficked via the plasma membrane, and then internalized and transported from the endosomal compartment to the lysosomes in an AP-3–dependent manner (Peden et al, 2004). In conditions of AP-3 deficiency, Lamp1 is recycled from the endosomes back to the plasma membrane, leading to its accumulation at the cell surface (Dell'Angelica et al, 1999; Peden et al, 2002). Immunostaining for surface and total Lamp1 demonstrated significant mislocalization of the protein to the plasma membrane in both β4-mCherry and β23-mCherry neurons (Fig 8C and D), suggesting that AP-3–dependent transport of lysosomal proteins is impaired. Total Lamp1 levels were not changed in the proteome of β protein–expressing neurons (Table S4 and Fig S7G).

Mice with a homozygous null mutation in AP-3δ1 (mocha mice) exhibit defects in lysosome-related organelles such as melanosomes and platelet dense granules and show altered secretion of lysosomal enzymes in the kidney, suggesting that AP-3 loss-of-function causes impairments in lysosomes (Swank et al, 1991). We sought to confirm this in a mocha fibroblast cell line, and used functionally rescued mocha cells stably expressing the AP-3δ1 subunit as control (Kent et al, 2012). In agreement with previous reports (Kantheti et al, 1998), the levels of AP-3μ1 were severely reduced in mocha cells (Fig S8A–C), consistent with the instability of the whole complex in the absence of one subunit. When mocha cells were loaded with LysoTracker Red, we observed reduced numbers of lysosomes per cell (Fig 8E and F), in line with our findings in LysoTracker-labeled β protein neurons (Fig 4C). These results confirm that impairment of the AP-3 complex leads to lysosomal defects.

To test whether re-supplying AP-3μ1 would be sufficient to ameliorate β protein toxicity, we quantified cell death in neuronal cultures co-transfected with β proteins and AP-3μ1 or EGFP as a control. Cleaved caspase-3 staining demonstrated that co-expression of AP-3μ1 significantly reduced cell death in both β4-mCherry and β23-mCherry cells (Fig S8D and E). Therefore, impaired integrity and function of the AP-3 adaptor complex contributes to β protein toxicity. Altogether, our data strongly suggest that lysosomal defects mediate, at least in part, the toxicity of β protein aggregates in neurons.

## Discussion

Here we used rationally designed aggregating proteins to understand the role of the aggregates' gain-of-function toxicity in neurons. It should be kept in mind that β proteins differ from natural amyloids in several aspects. The solvent-exposed and buried face of the rationally designed proteins consist entirely of polar and non-polar amino acids, respectively, whereas natural amyloids present a more mixed structure. In addition, natural amyloid fibrils are wider and more resistant to disassembly than those made up of β proteins

acidic organelles are indicated with white and blue boxes, respectively, and shown at higher magnification in the insets. **(F)** Quantification of the total number of reporter puncta per cell (n = 4 independent experiments; 15–45 cells/condition/experiment; two-tailed t test with Welch correction). **(G)** Quantification of the fraction of non-acidic (yellow) and acidic (red only) puncta. Same cells were analyzed as in (F) (two-way ANOVA with Sidak's post hoc test). **(H)** Western blots for early autophagy markers in lysates of HeLa cells transfected with mCherry or β23-mCherry. HPRT was used as a loading control. **(I)** Western blot quantification (n = 5 independent experiments; two-tailed t test). **(J)** Transcript levels of lysosomal genes in mCherry or β23-mCherry HeLa cells determined with RT-PCR (n = 6 experiments; two-tailed t test). Scale bar in (E), 5 μm. Data information: Data are presented as mean ± SD. *P < 0.05; **P < 0.01; ***P < 0.001; n.s., not significant.

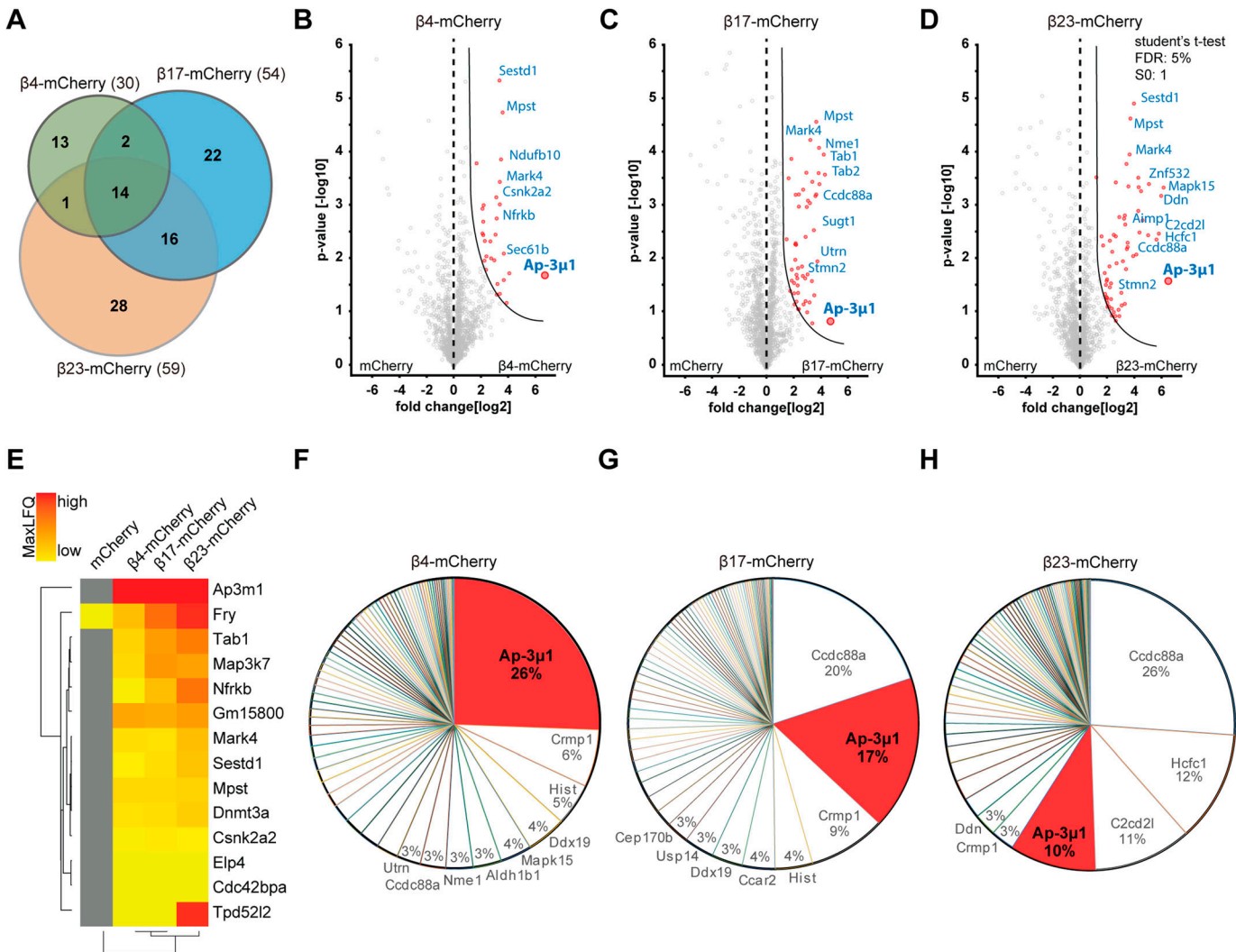

**Figure 6.  β protein interactome in primary neurons.**
**(A)** Venn diagram depicting numbers and overlap of interactors for the three β proteins investigated. **(B, C, D)** Volcano plots depicting proteins significantly enriched in β protein immunoprecipitates. Red dots denote proteins that pass 5% permutation-based false discovery rate (curved line on the right; proteins significantly associated with mCherry are not highlighted for the interactomes). **(E)** Heat map of common interactors of all three β proteins. **(F, G, H)** Pie charts showing the quantitative composition of β protein–interacting complexes.

(West et al, 1999; Olzscha et al, 2011; Fitzpatrick et al, 2017; Falcon et al, 2018; Zhang et al, 2020). Despite these limitations, β proteins have proven very useful in unraveling general mechanisms of aggregate toxicity that apply to natural disease-related proteins (Olzscha et al, 2011; Woerner et al, 2016; Vincenz-Donnelly et al, 2018; Frottin et al, 2019). Apart from fibrillar aggregates, soluble oligomers may also play a role in the effects observed here, as a large fraction of the β proteins remained soluble at the time point of the analyses (Fig S1).

Our integrated structural, functional, and molecular approach demonstrates that gain-of-function toxicity of aggregating proteins can be sufficient to cause lysosomal defects and neurotoxicity. These findings address a long-standing question regarding the role of aggregates as causative agents versus incidental by-products of disease progression (Haass & Selkoe, 2007; Arrasate & Finkbeiner, 2012) and conclusively show that protein aggregation per se has a deleterious impact on neurons. The toxicity mechanisms triggered by

aggregating proteins are multifactorial and involve various cellular processes (Olzscha et al, 2011; Kim et al, 2016). Although our results are in agreement with this view, they point to lysosomal degradation as a pathway that is particularly vulnerable in primary neurons.

The lysosomal abnormalities that we observed in β protein neurons are reminiscent of those reported in common neurodegenerative diseases such as Alzheimer's and Parkinson's (Suzuki & Terry, 1967; Nixon et al, 2005; Dehay et al, 2012; Usenovic et al, 2012; Gowrishankar et al, 2015). One possible cause of lysosomal defects in protein misfolding disorders is that protein aggregates are targeted by the autophagy-lysosomal pathway (Filimonenko et al, 2010; Jo et al, 2014; Menzies et al, 2015; Hoffmann et al, 2019), but fail to be degraded by lysosomes, accumulating in these organelles and impairing their function (Lamark & Johansen, 2012). In the case of β proteins, this scenario seems unlikely because no aggregates were found within lysosomes by cryo-ET or light microscopy.

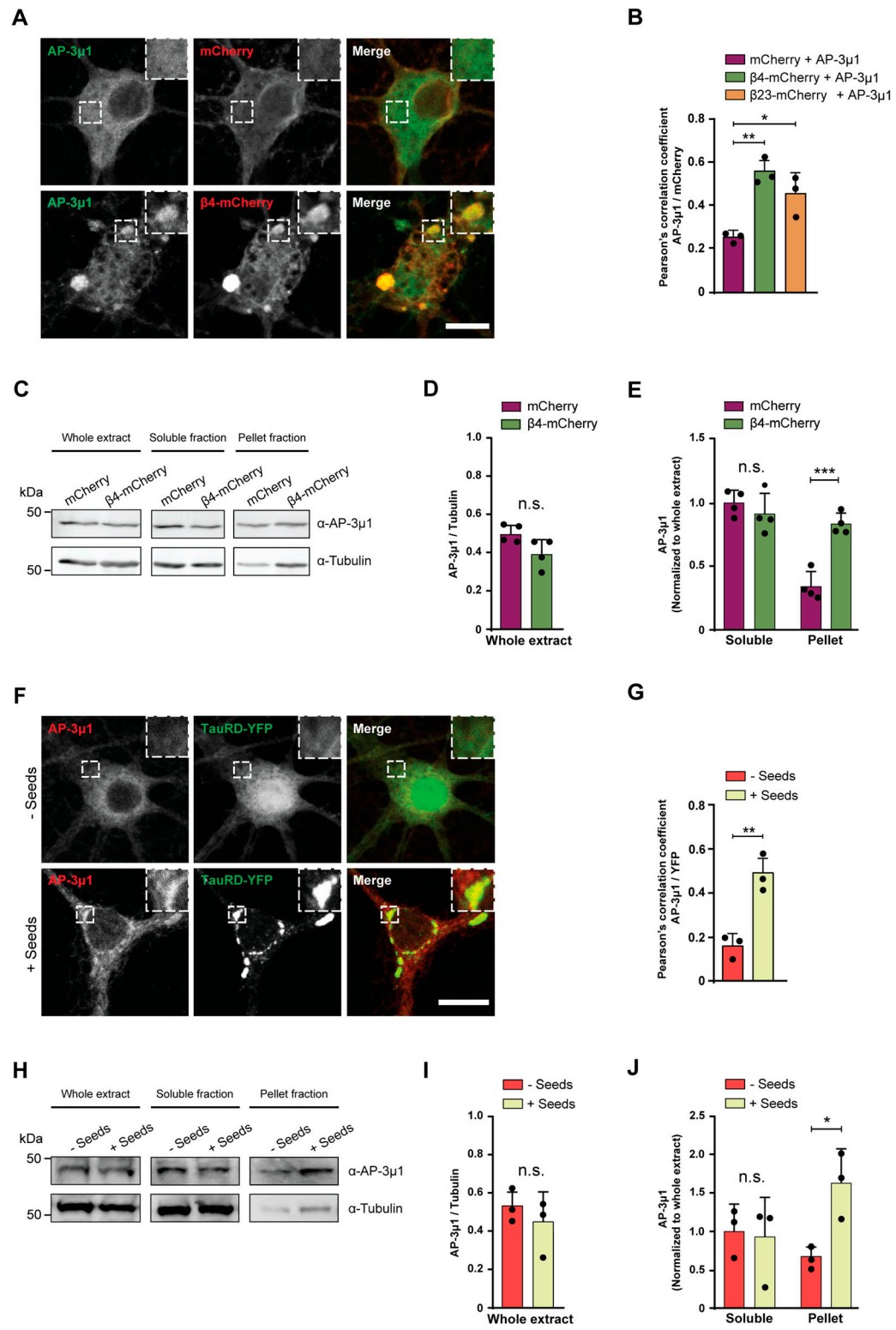

**Figure 7. AP-3µ1 is sequestered by aggregates, leading to defects of AP-3 complex.**
**(A)** Single confocal plane images of DIV 10+2 cortical neurons co-transfected with Flag-AP-3µ1 and either mCherry (top), or β4-mCherry (bottom). Anti-Flag staining was used to detect AP-3µ1. Areas marked by the boxes are magnified in the insets. **(B)** Quantification of Pearson's correlation coefficient between the AP-3µ1 and mCherry signal (n = 3 independent experiments, 20–30 cells/condition/experiment; one-way ANOVA with Tukey's post hoc test). **(C)** Western blots for AP-3µ1 in whole lysates (left), Triton X–soluble fraction (middle), and pellet fraction (right) of DIV 10+6 primary neurons transduced with mCherry or β4-mCherry. Tubulin was used as a loading control. **(D, E)** Quantification of (C). Levels of AP-3µ1 in the whole extract were normalized to tubulin. Levels of AP-3µ1 in the soluble and pellet fractions were normalized

We propose that impaired trafficking of lysosomal proteins due to an insufficient cellular pool of intact AP-3 complex contributes to the lysosomal defects caused by β protein aggregates. In agreement with previous studies investigating mutations in other AP-3 subunits (Peden et al, 2002), we show that reduced levels of AP-3μ1 compromise the stability of the whole complex. In turn, AP-3 defects lead to missorting of lysosomal proteins and impair lysosomal function. AP-3 has been implicated in protein trafficking at different subcellular locations, such as direct transport from the Golgi apparatus or internalization from the plasma membrane via the endosomal pathway (Dell'Angelica et al, 1999; Peden et al, 2004; Li et al, 2016). In neurons, the AP-3 complex is also involved in the polarized sorting of proteins to the axonal compartment (Li et al, 2016). Moreover, a number of other proteins related to intracellular transport and to other functions interact with β proteins. Thus, the exact trafficking step(s) impaired by AP-3 deficiency in aggregate-bearing neurons remain to be elucidated. It is also possible that impairment of cellular processes beyond lysosomal trafficking contributes to the deleterious effects of β proteins.

Importantly, we show that AP-3μ1 also accumulates in Tau aggregates. Moreover, previous proteomic studies demonstrated interactions of other AP-3 subunits with disease-related aggregating proteins. Thus, AP-3β2 was found to be an interactor of soluble Huntingtin (Shirasaki et al, 2012), and together with AP-3δ1 it was also identified as a component of insoluble mutant Huntingtin inclusions in the mouse brain (Hosp et al, 2017). AP-3β2 was among the most significant interactors of phosphorylated tau in human Alzheimer's disease brains (Drummond et al, 2020). Sequestration of other lysosomal proteins by aggregates and impaired trafficking of lysosomal enzymes was also described in synucleinopathies and in spinal and bulbar muscular atrophy (Chu et al, 2009; Dehay et al, 2012; Decressac et al, 2013; Cortes et al, 2014; Mazzulli et al, 2016). Therefore, compromised intracellular transport of lysosomal components may be an important factor in the pathogenesis of multiple neurodegenerative diseases.

## Materials and Methods

### Plasmids

For transfection, myc-β4 and myc-β23 (kind gift from Mark S Hipp and F Ulrich Hartl) were cloned into pmCherry-N1 (Clontech) between KpnI and AgeI restriction sites. pmCherry-N1 was used as control. The mouse AP-3μ1 expression plasmid was purchased from OriGene (Ref. MR206629) and consists of AP-3μ1-myc-DDK in pCMV6-ENTRY. For interactome analysis, myc-mCherry, myc-β4-mCherry, myc-β17-mCherry, and myc-β23-mCherry were cloned into pENTR1A-164

between restriction sites NotI and EcoRI. For β protein characterization in lentivirally transduced neurons, myc-mCherry and myc-β23-mCherry were cloned between BamHI and XhoI restriction sites of the plasmid pFhSynW2, whereas myc-β4-mCherry was cloned between BamHI and EcoRV. Plasmids for lentiviral expression included the Kozak sequence CCCACC at the 5′ terminus of the insert. For lentiviral TauRD-EYFP expression, pFhSynW2 TauRD (P301L/V337M)-EYFP was constructed by PCR amplification of the TauRD (P301L/V337M)-EYFP sequence from the N1-TauRD (P301L/V337M)-EYFP plasmid (gift from Marc Diamond). The PCR product was then digested and subcloned into the pFhSynW2 using XhoI and NheI restriction sites (Yuste-Checa et al, 2021).

### Lentivirus production

Hek293T cells for lentiviral packaging were purchased (Lenti-X 293T cell line; Takara Bio) and expanded to 70–85% confluency in DMEM Glutamax (+4.5 g/L D-Glucose, - Pyruvate) supplemented with 10% FBS (Sigma-Aldrich), 1% G418 (Gibco), 1% NEAA (Thermo Fisher Scientific), and 1% Hepes (Biomol). Only low-passage cells were used. For lentiviral production, three T75 cm$^2$ flasks (Falcon) containing 5.7 million cells each were seeded and henceforth cultured in medium without G418. On the following day, cells were transfected with the expression plasmid pFhSynW2, and the packaging plasmids psPAX2 and pVsVg (all three kindly provided by Dieter Edbauer) using TransIT-Lenti transfection reagent (Mirus Bio). The transfection mix was incubated for 20 min at RT and in the meanwhile, cell medium was exchanged. 1 ml transfection mix was added to each flask and left overnight. The medium was exchanged on the next day. After 48–52 h, culture medium containing the viral particles was collected and centrifuged for 10 min at 1,200g. Then, the supernatant was filtered through 0.45-μm-pore size filters using 50-ml syringes, transferred to Ultra-clear centrifuge tubes (Beckman Coulter), and centrifuged at 100,000g for 2 h in the Avanti JXN-30 centrifuge, rotor JS-24.38 (Beckman Coulter). Supernatant was discarded and the lentivirus pellet resuspended in TBS-5 buffer (50 mM Tris–HCl, pH 7.8, 130 mM NaCl, 10 mM KCl, and 5 mM MgCl$_2$). After aliquoting, virus was stored at −80°C. For interactome analyses, lentivirus was produced using the pLV-Syn vector (derived from pLenti7.3-V5-Dest) and the packaging vectors pCMVdelta8.91 and pMD2_G (kindly provided by Alexandra Lepier).

### Preparation of EM grids

R 2/1 Holey Carbon Au 200 mesh EM grids (Quantifoil) were coated with ~20 nm of carbon in a MED 020 carbon coater (BAL-TEC) and made hydrophilic for 45 s in a PDC-3XG plasma cleaner (Harrick Scientific Products). For HeLa cell culture, the grids were

---

to its levels in the whole extract (n = 4 independent cultures; (D), two-tailed t test with Welch's correction; (E), two-way ANOVA with Sidak's post hoc test). **(F)** Single confocal plane images of DIV 10+7 cortical neurons transduced with TauRD-EYFP, and treated with TauRD seeds (bottom) or PBS (top) on DIV 10+3. Areas marked by the boxes are magnified in the insets. **(G)** Quantification of Pearson's correlation coefficient between endogenous AP-3μ1 and TauRD-EYFP (n = 3 independent experiments, 20–30 cells/condition/experiment; two-tailed t test with Welch's correction). **(H)** Western blots for AP-3μ1 in whole lysates (left), Triton X–soluble fraction (middle) and pellet fraction (right) of DIV 10+7 cortical neurons transduced with TauRD-EYFP and treated or not treated with TauRD seeds. Tubulin was used as a loading control. **(I, J)** Quantification of (H). Levels of AP-3μ1 in the whole extract were normalized to tubulin. Levels of AP-3μ1 in the soluble and pellet fractions were normalized to its levels in the whole extract (n = 3 independent cultures; (I), two-tailed t test with Welch's correction; (J), two-way ANOVA with Sidak's post hoc test). Scale bars in (A, F), 10 μm. Data information: Data are presented as mean ± SD. *P < 0.05, **P < 0.01, ***P < 0.001, n.s., not significant.

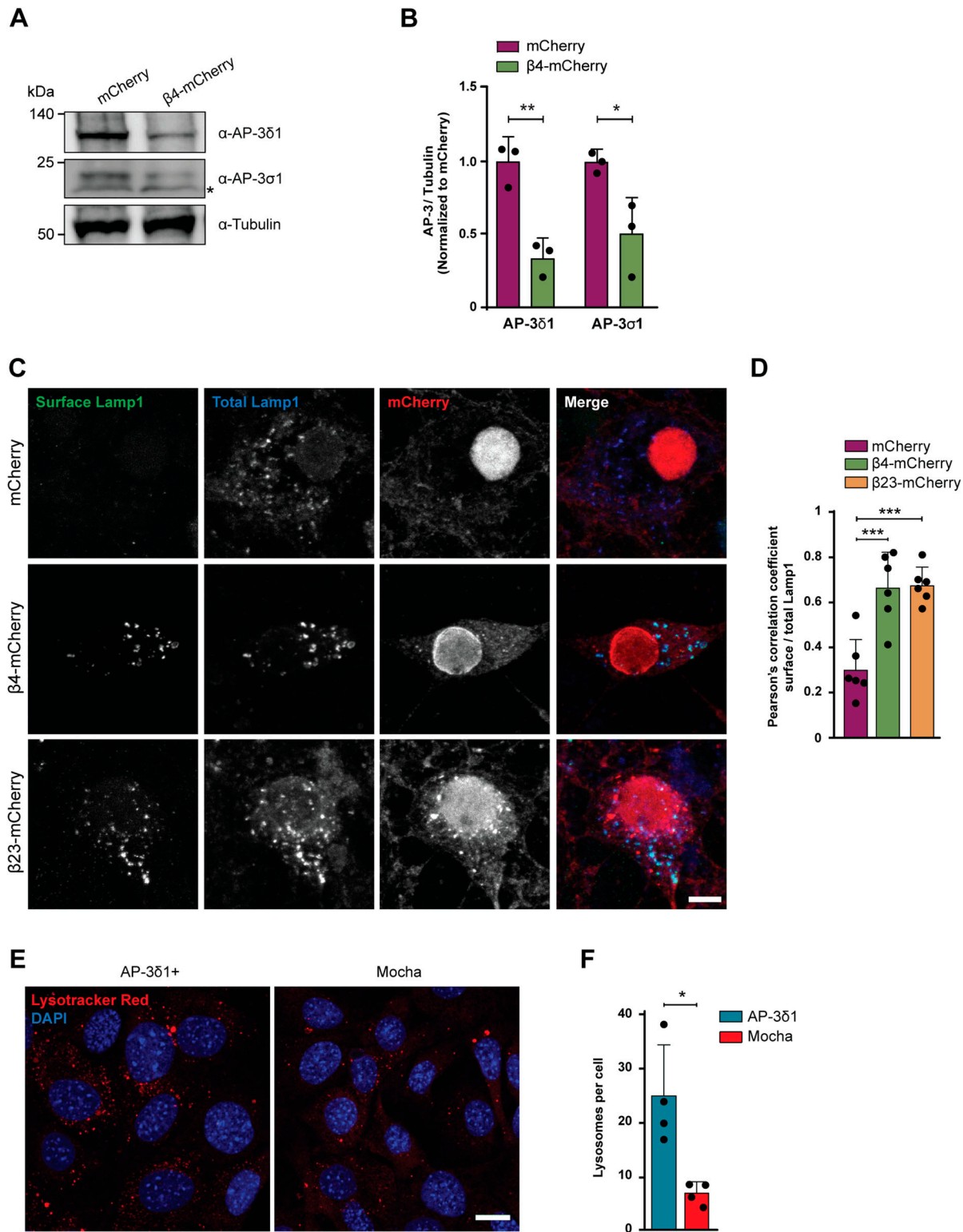

**Figure 8. Dysfunction of the AP-3 complex causes missorting of Lamp1 and lysosomal defects.**
**(A)** Western blots for AP-3δ1 and AP-3σ1 in whole lysates of DIV 10+6 primary neurons transduced with mCherry or β4-mCherry. Asterisk marks an unspecific band. Tubulin was used as a loading control. **(B)** Quantification of (A) (n = 3 independent cultures; two-tailed t test with Welch's correction). **(C)** Maximum intensity projections of DIV 10+4 neurons transduced with mCherry (top), β4-mCherry (middle), or β23-mCherry (bottom) and stained for surface (green) and total (blue) Lamp1. **(D)** Quantification of Pearson's correlation coefficient between surface and total Lamp1 signal (n = 6 independent experiments at DIV 10+4 and DIV 10+5, 25 cells/condition/experiment; one-way ANOVA with Dunnett's post hoc test). **(E)** Fluorescent images of control (left) and mocha cells (right) incubated with LysoTracker Red. Nuclei were labelled with DAPI (blue). **(F)** Quantification of lysosome numbers per cell (n = 4 independent experiments; at least 100 cells/condition/experiment; two-tailed t test with Welch's correction). Scale bars; (C), 10 μm; (E), 20 μm. Data information: Data are presented as mean ± SD. *P < 0.05, **P < 0.01, ***P < 0.001.

UV-sterilized in a Herasafe HR18 cell culture hood (Heraeus) for 30 min. For neuronal cultures, the grids were sterilized in ethanol for 10 min, washed several times in double-distilled water, and transferred to culture dishes containing water. Grids and dishes for neuronal cultures were coated with poly-D-lysine (Sigma-Aldrich; 1 mg/ml in borate buffer) for 24 h and washed three times with water. Subsequently, the grids were coated with laminin (Thermo Fisher Scientific; 5.0 µg/ml in PBS) for 24 h, washed with PBS three times, and placed in Neurobasal medium supplemented with B27 containing 0.5 mM Glutamine (all reagents from Thermo Fisher Scientific). During washes and medium exchange steps, grids were transferred into another dish containing the appropriate liquid to prevent them from drying.

### Primary neuron culture

Primary neurons were prepared from E14.5-15.5 CD-1 or C57BL/6 wild-type embryos. All experiments involving mice were performed in accordance with the relevant guidelines and regulations of the Government of Upper Bavaria. Pregnant mice were euthanized by cervical dislocation, the uterus was removed from the abdominal cavity and placed into a 10-cm sterile Petri dish on ice containing dissection medium, consisting of HBSS supplemented with 0.01 M HEPES, 0.01 M $MgSO_4$, and 1% Penicillin/Streptomycin. Each embryo was isolated, heads were quickly cut, brains were removed from the skull, and immersed in ice-cold dissection medium. Subsequently, cortical hemispheres were dissected and meninges were removed under a stereo-microscope. For each sample, cortical or hippocampal tissue from typically six to seven embryos was transferred to a 15-ml sterile tube and digested with 0.25% trypsin containing 1 mM 2,2′,2″,2′′′-(ethane-1,2-diyldinitrilo) tetraacetic acid (EDTA) and 15 µl 0.1% DNAse I for 20 min at 37°C. The enzymatic digestion was stopped by removing the supernatant and washing the tissue twice with Neurobasal medium (Invitrogen) containing 5% FBS. The tissue was resuspended in 2 ml medium and triturated to achieve a single-cell suspension. Cells were spun at 130–180$g$, the supernatant was removed, and the cell pellet was resuspended in Neurobasal medium with 2% B27 (Invitrogen), 1% L-glutamine (Invitrogen) and 1% penicillin/streptomycin (Invitrogen). For cryo-ET, neurons were plated on the coated grids within 24-well plates (60,000 per well). For MS analysis, cells were cultured on six-well plates (Thermo Fisher Scientific) (500,000 neurons per well) coated with 1 mg/ml poly-D-lysine (Sigma-Aldrich) and 1 µg/ml laminin (Thermo Fisher Scientific). For immunostaining, neurons were cultured on 13-mm coverslips, coated as above, in 24-well plates (Thermo Fisher Scientific). For MTT assay, neurons were cultured in coated 96-well plates. Transfection was performed using Calcium Phosphate according to the protocol from Jiang and Chen (2006) or using Lipofectamine 2000 (Thermo Fisher Scientific) according to the manufacturer's protocol. For lentiviral transduction, viruses were thawed and immediately added to freshly prepared neuronal culture medium. A fifth of the medium from cultured neurons was removed and the equivalent volume of virus-containing medium was added. Typically, 1 µl/cm$^2$ of virus was added, but this amount was sometimes adjusted to match protein expression levels among different constructs. TauRD (residues 244–371, C291A/P301L/C322A/V337M) seeds were kind gift of Patricia Yuste-Checa and were prepared as described (Yuste-Checa et al, 2021). Seeding was performed on DIV 10+3 by adding 70 ng of TauRD seeds mixed with fresh medium (1–10th of the medium in the well) to the neuronal cultures in 24-well plates.

### HeLa cell culture

HeLa CCL-2 (ATCC) cells were grown in DMEM medium (Life Technologies) with 10% FBS (Life Technologies), 0.5 mM L-glutamine (Life Technologies), 1% MEM NEAA 100× (Life Technologies), and 1% penicillin/streptomycin (10,000 U/ml; Life Technologies) at 37°C and 5% $CO_2$. HeLa cells were transfected using FuGENE 6 transfection reagent (Promega) according to the manufacturer's instructions. For cryo-ET, ~25,000 cells were seeded in 35-mm culture dishes (BD Falcon) containing four pre-treated EM grids 24 h before transfection. For Western blot experiments, cells were plated in 6 cm dishes 24 h before transfection (250,000 cells/well). To inhibit autophagy, cells were treated with 50 µM chloroquine in water (Sigma-Aldrich). HeLa cells were harvested 24–48 h after transfection.

### Mocha cell culture

Mocha cells and control cells stably expressing AP-3δ1 (AP-3δ1+ cells) were kind gift of Andrew A Peden. Cells were cultured in DMEM supplemented with 10% FCS and 1% penicillin/streptomycin. For AP-3δ1+ cells, 0.2 mg/ml hygromycin B was added to the medium. For lysosome counting, 100,000 cells were seeded onto glass coverslips. Cells were treated with 150 nM LysoTracker Red (Thermo Fisher Scientific) for 30 min 1 d after seeding, washed with PBS and fixed with 4% PFA. Lysosomes were automatically quantified using a custom-made CellProfiler pipeline. For Western blot, 250,000 cells were seeded in six-well plates and harvested 1 d after seeding.

### Immunostaining

Cells were fixed with 4% PFA in PBS for 20 min; remaining free groups of PFA were blocked with 50 mM ammonium chloride in PBS for 10 min at RT. Cells were rinsed once with PBS and permeabilized with 0.25% Triton X-100 in PBS for 5 min. After permeabilization, cells were washed with PBS and blocked with 2% BSA (wt/vol) (Carl Roth) and 4% donkey serum (vol/vol) (Jackson Immunoresearch Laboratories) in PBS for 30 min at RT. Coverslips were transferred to a light-protected humid chamber and incubated in primary antibody diluted in blocking solution for 1 h. The following primary antibodies were used: anti-mCherry (AB0040-200; OriGene, 1:500), anti-myc 9E10 (13-2500; Thermo Fisher Scientific, 1:100), anti-cleaved caspase-3 (9661S; Cell Signaling Technology, 1:500), anti-Flag (TA-50011-100; OriGene, 1:500), and anti–AP-3µ1 (ab201227; Abcam, 1:300). Cells were washed with PBS and incubated with secondary antibodies (Jackson Immunoresearch Laboratories) diluted 1:250 in blocking solution, with 0.5 µg/ml DAPI added to stain the nuclei. Coverslips were mounted on Menzer glass slides using Dako or Prolong Glass fluorescence mounting medium (Thermo Fisher Scientific). Confocal images were obtained at a spinning disc microscope (Zeiss) or SP8 confocal microscope (Leica).

Colocalization analysis for AP-3$\mu$1 and $\beta$-mCherry proteins/TauRD-EYFP was performed on single plane images. A regions of interest (ROI) was manually drawn around the soma of each cell in the mCherry/EYFP channel and the ImageJ plug-in Coloc 2 was used to calculate the Pearson's correlation coefficient.

### Lamp1 staining

Transduced cortical and hippocampal neurons were used. Stainings were performed at DIV 10+4 and 10+5 (three cultures for each time point) as described (Glynn & McAllister, 2006). Briefly, live neurons cultured on coverslips were washed with ice-cold PBS and placed in a pre-cooled humid dark chamber that was kept on ice. Primary antibody (rat $\alpha$-Lamp1; Abcam, Reference ab25245) diluted 1:300 in blocking solution was added for 1 h. After washing with ice-cold PBS, secondary antibody (donkey $\alpha$-rat 488) diluted 1:250 was added for 30 min. Cells were washed with ice-cold PBS and fixed with ice-cold 4% PFA in PBS for 10 min at RT. Next steps were performed as described above in Immunostaining, using the same primary antibody, but changing the secondary to donkey $\alpha$-rat 647 to differentiate between surface and total Lamp1.

Colocalization analysis was performed on maximum intensity projection images of transduced neurons. An ROI was manually drawn around the soma of each cell in the mCherry channel. A background subtraction of rolling ball radius of 50 pixels was performed in all images. The ImageJ plug-in Coloc 2 was used to calculate the Pearson's correlation coefficient between surface and total Lamp1.

### Neuronal viability measurements and Sholl analysis

For transduced neurons, viability was determined with the MTT assay using Thiazolyl Blue Tetrazolium Bromide (MTT) reagent purchased from Sigma-Aldrich. Neurons were cultured in 96-well plates. First, cell medium was exchanged for 100 $\mu$l of fresh medium. Then, 20 $\mu$l of 5 mg/ml MTT in PBS was added and incubated for 2–4 h at 37°C, 5% $CO_2$. Subsequently, 100 $\mu$l solubilizer solution (10% SDS, 45% dimethylformamide in water, pH 4.5) was added, and on the following day, absorbance was measured at 570 nm. Each condition was measured in triplicates and absorbance values averaged for each experiment.

For transfected neurons, toxicity was quantified by an investigator blinded to the conditions based on immunostaining for cleaved caspase-3.

For Sholl analysis, mCherry channel was used to visualize neuronal morphology. Coverslips were immunostained against mCherry and imaged with a 40× objective. Maximum intensity projections were analyzed. First, cell morphologies were semiautomatically traced with the Simple Neurite Tracer plug-in of ImageJ in a blinded way. Then, complexity of the traced neurons was quantified with the Sholl analysis plug-in of ImageJ using a custom-made macro for batch processing.

### Live cell light microscopy

Samples were imaged using bright-field and epifluorescence wide-field microscopy in a CorrSight microscope (FEI) using FEI MAPS 2.1 software. Low magnification overview images were acquired with a 5× air objective (Plan-ApoChromat, NA 0.16; Carl Zeiss). For high magnification image stacks, a 63× oil immersion objective (Plan Achromat, NA 1.4; Carl Zeiss) was used. Cells were kept on a 37°C heated stage in a homemade climate chamber infused with humidified air and 5% $CO_2$ gas. Cells were located at low magnification and z-stacks were acquired in 500 nm steps over the whole cell height at high magnification.

For the analysis of lysosome size and number, neuronal cultures expressing $\beta$23-mCherry, $\beta$4-mCherry, or mCherry were incubated with 75 nM LysoTracker Green DND-26 (Thermo Fisher Scientific) for 30 min according to the manufacturer's instructions. The mCherry signal was used to assess the cell perimeter and only lysosomes within that perimeter were measured. The mCherry signal of each cell was isolated by automatic thresholding using a homemade script written for Fiji software (Schindelin et al, 2012). The size of each lysosome was manually measured in its in-focus plane using Fiji software. The Pearson correlation coefficient was measured using JACoP plugin in Fiji software (Bolte & Cordelieres, 2006).

For autophagy flux analysis with mCherry-GFP-LC3 reporter, primary cortical neurons were seeded in imaging plates with a glass-bottom (Nunc) and co-transfected at DIV 10 with the mCherry-GFP-LC3 reporter (gift from Anne Brunet, plasmid # 110060; Addgene) and either $\alpha$-S824 or myc-$\beta$4 protein. Live imaging was performed at DIV 10+3 using a Spinning Disc Microscope SN 3834000224 (Zeiss) and the Visiview Software (Visitron Systems). Single imaging planes of each neuron were selected for quantification. The total number of puncta and the number of yellow puncta were manually quantified for each neuron. The number of red puncta was calculated by subtracting the number of yellow puncta from the total. All quantifications were performed in a blinded fashion to the analyzed condition.

### Sample vitrification

Cells were vitrified using a homemade manual gravity-driven plunge freezer. Before plunge-freezing, cells were treated with 10% glycerol (Sigma-Aldrich) in medium as a cryoprotectant for 1–5 min. In addition, in some cases, cells were stained with 3% Annexin V and Alexa Fluor 488 conjugate (Life Technologies) 15 min before vitrification to identify dead cells. During plunge freezing, EM grids were blotted for 8 s with filter paper No. 1 (Whatman) from the back side and immediately frozen in liquid ethane/propane (37%/63%; Linde) kept at liquid nitrogen temperature. Grids were transferred to liquid nitrogen and excess ethane/propane was blotted with pre-cooled filter paper. Grids were stored in liquid nitrogen until further use.

### Cryo-light microscopy

Frozen grids were fixed into FIB C-clip rings to increase mechanical stability. Samples were transferred to a CorrSight shuttle (FEI) and mounted on the CorrSight cryo-stage (FEI) maintained at liquid nitrogen temperature for cryo-light microscopy. Samples were imaged using wide-field or spinning-disk confocal, epifluorescence microscopy using FEI MAPS 2.1 software. Images were acquired with a 1,344 × 1,024 pixel camera (pixel size 6.4 $\mu$m, Hamamatsu Digital

Camera C10600 ORCA-R2). Grids were imaged in wide-field mode at low magnification with a 5× air objective (Plan-ApoChromat, NA 0.16; Carl Zeiss) and in spinning-disk confocal mode at high magnification with a 40× air objective (EC Plan-Neofluar, NA 0.9; Carl Zeiss) for identification of cells and/or aggregates. Image acquisition and further SEM correlation was performed by three-point correlation using FEI MAPS 2.1 software.

### Cryo FIB SEM

Grids imaged by light microscopy were mounted on a transfer shuttle designed for a cryo-loading system (Rigort et al, 2010) and loaded into a Quanta 3D FEG dual-beam FIB/SEM (FEI). Grids were sputtered with platinum (10 mA, 30 s) in a PP3000T loading system (Quorum) to reduce charging effects during electron imaging. Grids were loaded into the FIB chamber and coated with organometallic platinum as a protective layer for ion beam milling. Grids were imaged with the scanning electron beam operated at 5 kV/12 pA and ROI were identified via cryo-LM/SEM three-point correlation using MAPS 2.1 software. ROIs were thinned down at tilt angles of 18°–20° with the focused ion beam operated at 30 kV. The beam currents where set to 1 nA at ~1 $\mu$m distance from the ROI, 500 pA at 750 nm, 300 pA at 400 nm, 100 pA at 250 nm, 50 pA at 100 nm, and 30 pA at 75 nm for polishing. Grids were sputtered once more with platinum (10 mA, 5 s) after milling to increase conductivity of the lamellae for VPP imaging.

### Cryo-ET

Cryo-TEM was performed on a Titan Krios cryo-TEM (FEI) operated at 300 kV, equipped with a FEG, post-column energy filter (Gatan) and Volta phase plate (Danev et al, 2014). Tomograms were acquired on a K2 Summit direct electron detector (Gatan) in dose fractionation mode (0.08 frames per second) using SerialEM (Mastronarde, 2005). Lamella overview images were acquired at 3,600× magnification (pixel size: 38.93 Å) and stitched using the automatic photo merge in Adobe Photoshop. For cryo-LM − cryo-TEM correlation, stitched lamella TEM overview images were imported into MAPS and aligned with cryo-LM data. Tilt series were recorded at −0.5 $\mu$m defocus for VPP imaging at 33,000× magnification (pixel size: 4.21 Å), with an angular increment of 2° and typically ranged from −50° to 60°. The total dose was restricted to ~120 electrons/Å$^2$ per tomogram.

### Tomogram reconstruction and data processing

K2 frames were aligned and combined using in-house software based on previous work (Li et al, 2013). Tilt series were aligned using patch tracking from the IMOD software package (Kremer et al, 1996) and reconstructed by weighted back projection. The platinum layer for VPP imaging and large pieces of surface ice contamination were computationally removed to increase signal-to-noise ratio (Fernandez et al, 2016). The resulting tilt series were realigned and reconstructed again. Tomograms were binned four times to a final pixel size of 16.84 Å to increase contrast.

Tomogram segmentation was performed using Amira (FEI). Membranes and microtubules were automatically segmented using TomoSegMemTV (Martinez-Sanchez et al, 2014) and corrected using

Amira. $\beta$-protein fibrils were traced on denoised tomograms (non-local means filter) by removing membranes and macromolecules followed by density thresholding and subsequent skeletonization. For the identification of ribosome positions, template matching was applied using the pyTOM toolbox in MATLAB (Mathworks) as previously reported (Hrabe et al, 2012). A human 80S ribosome structure (PDB: 4UG0) (Khatter et al, 2015) was filtered to 40 Å and used as a template. Coordinates of ribosomes identified via template matching were imported into Amira and masked with the filtered molecular structure. The diameter of the $\beta$ protein fibrils was measured in tomograms from neurons using IMOD.

### Western blotting

Cells were lysed in 50 mM Tris–HCl, pH 7.4, 150 mM NaCl, 2 mM EDTA, and 1% Triton X-100 with protease and phosphatase inhibitor tablets (Roche). Lysates were centrifuged for 10 min at 4,000$g$ (neurons) or 14,000$g$ (HeLa) and 4°C and supernatants were collected (soluble fraction). Pellets were used as insoluble fraction, whole extracts were used as total. Proteins were separated on 12% or 15% SDS–PAGE gels and transferred onto polyvinylidene difluoride (PVDF) membranes using a Trans-Blot Turbo transfer system (Bio-Rad). Blocking was performed with 3% BSA, 5% dried milk (Roth) in TBS-T for 1 h at RT. Primary antibodies were diluted 1:1,000 in 3% BSA in TBS-T, 0.01% sodium azide and incubated for 1 or 2 d at 4°C. Secondary antibodies were diluted 1:5,000 in 5% dried milk in TBS-T and applied for 2 h at RT. Tubulin-Rhodamine (Bio-Rad) was diluted 1:2,500 and added together with the secondary antibodies. Detection was performed using the Chemidoc MP imaging system (Bio-Rad). Primary antibodies used were anti-LC3B (D11) (3868; Cell Signaling Technology), anti-p62 (ab56416; Abcam), anti-GFP (632381; Clontech), hFAB rhodamine anti-tubulin (12004166; Bio-Rad), anti-myc 9E10 (13-2500; Thermo Fisher Scientific), anti-Atg5 (D5F5U; Cell Signaling Technology), anti-beclin-1 (3495; Cell Signaling Technology), anti-HPRT (ab10479; Abcam), anti–AP-3$\mu$1 (ab201227; Abcam), anti-AP-3$\delta$1 (kind gift from Andrew A Peden), anti-AP-3$\sigma$1 (TA319631; OriGene), and anti-tubulin (T9026; Sigma-Aldrich). For fluorescent Western blots, secondary antibodies were StarBright Blue 520 Goat Anti-Mouse, StarBright Blue 520 Goat Anti-Rabbit, StarBright Blue 700 Goat Anti-Mouse, and StarBright Blue 700 Goat Anti-Rabbit. For chemiluminescent Western blots, secondary antibodies were anti-rabbit HRP (NA934; Thermo Fisher Scientific) and anti-mouse HRP (NA931; Thermo Fisher Scientific). Band density quantification was performed with ImageJ and normalized to loading controls.

### Quantitative RT-PCR

RNA isolation and purification was performed using a Crystal RNA mini Kit (Biolab) according to the manufacturer's instructions. RNA quantification and quality control were performed with a NanoDrop spectrophotometer (PeqLab). cDNA was synthesized using a High-Capacity cDNA Reverse Transcription Kit (Applied Biosystems) according to the manufacturer's instructions. cDNA was diluted 1:50, each 8-$\mu$l reaction contained 4 $\mu$l diluted cDNA, 0.2 $\mu$l dilutions of each primer (25 $\mu$M stock), and 3.6 $\mu$l Luna Universal Probe qPCR

Master Mix (New England BioLabs). The primers were described previously (Fernandez-Mosquera et al, 2017).

## Immunoprecipitation

For AP-3μ1 immunoprecipitation, lysates of transduced cortical neurons cultured in six-well plates were incubated overnight with 2 μl of rabbit α-AP-3μ1 antibody (Reference ab201227; Abcam) on a rotating wheel at 4°C. 50 μl of Protein G Sepharose 4 Fast Flow (Sigma-Aldrich) slurry were washed, resuspended in lysis buffer (50 mM Tris–HCl, pH 7.4, 150 mM NaCl, 2 mM EDTA, and 1% Triton X-100 with protease and phosphatase inhibitor tablets [Roche]), and added to the lysates. The mix was incubated for 2 h at RT on a rotating wheel. Further steps were performed as described by the manufacturer, but performing all washes with lysis buffer.

For interactome analysis, cortical neurons (500,000 cells/condition) were harvested in ice-cold lysis buffer (0.25% NP40, 5% glycerol, 50 mM Tris HCl, and 150 mM NaCl) containing protease inhibitor (Roche), DNAse (5 μl/ml) and RNAse (0.5 μl/ml). Thermo Fisher Scientific Pierce Protein G Agarose (10 μl/100 μg IgG) was used for purification and immunoprecipitation of IgG antibodies. Serum samples containing IgG were incubated with Protein G agarose in a buffer that facilitates binding (20 mM sodium phosphate) for 1 h. Then, neuronal lysates were loaded onto IgG-protein G Agarose and incubated for 4 h. Non-IgG and non-antigen components were discarded from the sample by washing with 0.1% NP40, 5% glycerol, 50 mM Tris HCl, and 150 mM NaCl. Enriched proteins were next isolated and processed in a step-wise manner. First, proteins were denatured, digested, and cysteines reduced for 1 h at 20°C by adding 8 M urea, 40 mM Hepes, pH 8.0, 1 μg LysC, and 10 mM DTT. Next, urea concentration was lowered to 2 M by adding 50 mM ammonium bicarbonate in MilliQ water. Trypsin (1 μg) was added to the protein-peptide mix and samples were subjected to 55 mM chloroacetamide for 1 h to alkylate cysteine residues. Alkylation was quenched by adding 2 M urea/50 mM thiourea and overnight digestion was carried out after addition of 1 μg fresh trypsin. Interactome analysis was conducted in four independent biological replicates.

## Complete neuronal proteome precipitation

Complete neuronal proteomes were extracted as described previously (Hornburg et al, 2014). In brief, cells were lysed in lysis buffer (4% SDS and 10 mM Hepes, pH 8.0) and reduced with 10 mM DTT for 30 min. The proteins were then subjected to 45 min of alkylation with 55 mM iodoacetamide. Acetone precipitation was performed to remove the SDS. Acetone (−20°C) was added to 100 μg of proteins to a final concentration of 80% vol/vol, and proteins were precipitated overnight at −20°C. The supernatant was removed after 15 min of centrifugation (4°C, 16,000g) followed by washing with 80% acetone (−20°C). Residual acetone was evaporated at 20°C. The protein pellet was dissolved in 50 μl of 6 M urea/2 M thiourea and 10 mM Hepes, pH 8.0. LysC (1 μg), and digestion was carried out for 2 h at 20°C. The samples were incubated with 1 μg trypsin for overnight digestion and peptides were desalted on C18 solid phase extraction.

## Solid phase extraction

Stage Tips were prepared with 3xC18 material for rapid desalting and step elution of the peptide mixtures. Stage Tips were rinsed

with MeOH and buffer A (0.5% acetic acid). Samples were added to the staging tips and washed with buffer A. Buffer B (80% acetonitrile and 0.5% acetic acid) was used to elute the samples. Speedvac was used to remove the solvent from the samples. The samples were then resuspended in 10 μl buffer A* (0.5% AcOH, 0.1% TFA, and 2% ACN).

## LC-MS/MS

Peptides were separated on EASY-nLC 1000 HPLC system (Thermo Fisher Scientific). Columns (75-μm inner diameter, 40-cm length) were in-house packed with 1.9-μm C18 particles (Dr. Maisch GmbH, Ammerbuch-Entringen, Germany). Peptides were loaded in buffer A (0.5% formic acid) and separated with a gradient from 7% to 60% buffer B (80% acetonitrile, 0.5% formic acid) within 3 h 30 min at 200 nl/min. The column temperature was set to 60°C. A quadrupole Orbitrap mass spectrometer (Q Exactive; Thermo Fisher Scientific) was directly coupled to the liquid chromatograph via a nano-electrospray source. The Q Exactive was operated in a data-dependent mode. The survey scan range was set to 300–1,650 m/z, with a resolution of 70,000 at m/z 200. Up to 15 most abundant isotope patterns with a charge of ≥2 were subjected to higher energy collisional dissociation (Olsen et al, 2007) with a normalized collision energy of 25, an isolation window of 2 Th, and a resolution of 17,500 at m/z 200. To limit repeated sequencing, dynamic exclusion of sequenced peptides was set to 30 s. Thresholds for ion injection time and ion target value were set to 20 ms and $3 \times 10 \times 10^6$ for the survey scans and to 60 ms and $10 \times 10^6$ for the MS/MS scans. Data were acquired using Xcalibur software (Thermo Fisher Scientific).

## MS data analysis

To process MS raw files, we used MaxQuant software (v. 1.5.7.10) (Cox & Mann, 2008). Andromeda (Cox et al, 2011), which is integrated into MaxQuant, was used to search MS/MS spectra against the UniProtKB FASTA database. For the standard immunoprecipitation and pre-loaded serum, enzyme specificity was set to trypsin and LysC. For all the experiments, N-terminal cleavage to proline and up to two miscleavages was allowed. Peptides with a minimum length of seven amino acids were considered for identification. Oxidation, acetylation, and deamidation were set as variable modifications (maximum number of modifications per peptide was 5). A false discovery rate (FDR) cut-off of 1% was applied at the peptide and protein levels. Initial precursor mass deviation of up to 4.5 ppm and fragment mass deviation up to 20 ppm were allowed. Precursor ion mass accuracy was improved by time-dependent recalibration algorithms in MaxQuant. The cut-off score (delta score) for accepting individual MS/MS spectra was 17.

The proteome fasta file from *Mus musculus* (Taxon identifier: 10090) was used as a library for matches. Nonlinear retention time alignment (Cox & Mann, 2008; Cox et al, 2011) of all measured samples was performed in MaxQuant. "Match between runs," which allows the transfer of peptide identifications in the absence of sequencing, was enabled with a maximum retention time window of 0.7 min. Furthermore, we filtered our data by requiring a minimum peptide ratio count of 1 in MaxLFQ. Protein identification

required at least one razor peptide (Cox & Mann, 2008). Proteins that could not be discriminated on the basis of unique peptides were grouped into protein groups. Co-immunoprecipitated proteins were considered interactors if they were significantly enriched compared with the control samples applying a 5% permutation-based FDR cut-off (see below).

For statistical and bioinformatic analysis, as well as for visualization, we used the open Perseus environment, which is part of MaxQuant. The numerical data were transformed to $\log_2(x)$. Proteins were filtered for common contaminants; proteins identified only by site modification and reverse proteins were excluded. Missing values in pairwise comparisons were imputed with a normal distribution (width = 0.3 × SD; down shift = 1.8 × SD) in Perseus. Proteins were only considered if they were detected with at least three valid values among quadruplicates in at least one condition. For pairwise comparison of proteomes and determination of significant differences in protein abundances, *t* test statistics were applied with a permutation-based FDR of 5% and S0 of 1. The resulting significant outliers for each of the sample pairs were analyzed for gene ontology (GOCC) (Harris et al, 2004), biological ontology (GOBP), molecular function (GOMF), protein complexes (CORUM) (Ruepp et al, 2010), and protein families and domains (Pfam) (Finn et al, 2014) annotation enrichments. The data were displayed in a scatter plot for visual representation. For protein abundance ranking, summed peptide intensities from MaxQuant (median of at least three valid values within quadruplicates) were divided by the molecular weight to adjust for protein size bias and estimate the abundance rank of each protein within the proteome. $\log_{10}$ corrected intensities were plotted against the rank.

To calculate the content of disordered regions, we used R (rjson and seqinr libraries). First, amino acids that are predicted with low-complexity long region (IUPred-L) were mapped to sequences of proteins that are significant outliers in either of the β protein interactomes as well as for the entire population of proteins identified in the interactomes. Next, the ratio of all amino acids and those predicted with low complexity was calculated. To determine the significance of differences between the individual populations, a two-sample Wilcoxon test was performed on the ratio distributions.

### Aggregate composition estimation

To estimate the relative mass composition of the protein aggregates, the median of $\log_2$ MaxLFQ intensity (requiring at least two valid values) was calculated across quadruplicates for the β protein samples and the mCherry controls, respectively. Missing values in the mCherry samples were imputed with a normal distribution on the $\log_2$ scale (width = 0.3 × SD; down shift = 1.8 × SD). Next, the non-log (median) intensity of each protein in the β protein samples was subtracted by the respective non-log intensity derived from the mCherry samples. This transformation corrects for the overall background signal in the immunoprecipitation. The resulting intensities roughly correspond to the relative abundance in the aggregate (=composition) and are plotted as fractions in pie charts. Note that we did not apply an FDR or *P*-value cut-off for this analysis. By subtracting protein intensity of the mCherry controls from those in the β protein samples, small differences (which

associate with larger *P*-values and FDRs) have a diminishing contribution to the overall aggregate composition estimate. In the pie charts, proteins with at least 3% intensity contribution are listed.

### Statistical analysis

Statistical analysis of the MS data is described in the previous section. Statistical analysis of other data was performed using Origin Pro 2015G or GraphPad Prism 6. Data are presented as means ± SD unless indicated otherwise.

## Data Availability

The MS data from this publication are provided as supplemental tables. Raw data have been deposited to the PRIDE database (http://www.ebi.ac.uk/pride) and assigned the identifier PXD029689.

## Supplementary Information

## Acknowledgements

We thank F Ulrich Hartl, Mark S Hipp, Massimiliano Stagi, Georg HH Borner, and Shivani Tiwary for helpful discussions; Günter Pfeifer, Jürgen Plitzko, and Miroslava Schaffer for electron microscopy support; Qiang Guo for ribosome template matching; Andrew A Peden, F Ulrich Hartl, Mark S Hipp, Patricia Yuste-Checa, and Victoria A Trinkaus for sharing cell lines, plasmids, and reagents; Alexandra Lepier, Pontus Klein, Dieter Edbauer, and Carina Lehmer for lentiviral plasmids and generous help with lentivirus generation; Martin Dodel for excellent technical assistance; and Daniel del Toro Ruiz for kind help with image analysis. This work was funded by the European Research Council (ERC) Synergy Grant under FP7 GA number ERC-2012-SyG_318987-Toxic Protein Aggregation in Neurodegeneration (ToPAG) (to M Mann, W Baumeister, and R Klein); ERC Starting Grant MitoPexLysoNETWORK 337327 (to N Raimundo); Deutsche Forschungsgemeinschaft (DFG, German Research Foundation) through Germany's Excellence Strategy - EXC 2067/1-390729940 (to R Fernández-Busnadiego); SFB1286/A12 (to R Fernández-Busnadiego); DFG Project-ID 408885537 (TRR 274) (to F Meissner); the Horst Kübler-Stiftung (to I Dudanova); and by the Max Planck Society for the Advancement of Science.

### Author Contributions

I Riera-Tur: formal analysis, investigation, and visualization.
T Schaefer: formal analysis, investigation, and visualization.
D Hornburg: formal analysis, investigation, and visualization.
A Mishra: formal analysis and investigation.
M da Silva Padilha: formal analysis, investigation, and visualization.
L Fernández-Mosquera: formal analysis and investigation.
D Feigenbutz: investigation.
P Auer: investigation.
M Mann: conceptualization, supervision, and funding acquisition.
W Baumeister: conceptualization, supervision, and funding acquisition.
R Klein: conceptualization, supervision, and funding acquisition.

F Meissner: conceptualization, supervision, and project administration.
N Raimundo: conceptualization and supervision.
R Fernández-Busnadiego: conceptualization, supervision, project administration, and writing—original draft, review, and editing.
I Dudanova: conceptualization, supervision, visualization, project administration, and writing—original draft, review, and editing.

## Conflict of Interest Statement

The authors declare that they have no conflict of interest.

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
