## [Reviewer comments · Life Science Alliance]

Life Science Alliance

Amyloid-like aggregating proteins cause lysosomal defects in neurons via gain-of-function toxicity

Irene Riera-Tur, Tillman Schaefer, Daniel Hornburg, Archana Mishra, Miguel da Silva Padilha, Lorena Fernández-Mosquera, Dennis Feigenbutz, Patrick Auer, Matthias Mann, Wolfgang Baumeister, Rudiger Klein, Felix Meissner, Nuno Raimundo, Rubén Fernández-Busnadiego, and Irina Dudanova

DOI: <https://doi.org/10.26508/lsa.202101185>

Corresponding author(s): Irina Dudanova, Max Planck Institute of Neurobiology and Rubén Fernández-Busnadiego, Max Planck Institute of Biochemistry

Review Timeline:

Submission Date:	2021-08-05
Editorial Decision:	2021-08-05
Revision Received:	2021-11-12
Editorial Decision:	2021-11-29
Revision Received:	2021-12-02
Accepted:	2021-12-03

Transaction Report:

Please note that the manuscript was reviewed at *Review Commons* and these reports were taken into account in the decision-making process at *Life Science Alliance*.

Review
COMMONS

August 5, 2021

Re: Life Science Alliance manuscript #LSA-2021-01185

Dr. Irina Dudanova
Max Planck Institute of Neurobiology
Am Klopferspitz 18
Martinsried 82152
Germany [DE]

Dear Dr. Dudanova,

Thank you for submitting your manuscript entitled "Amyloid-like aggregating proteins cause lysosomal defects in neurons via gain-of-function toxicity" to Life Science Alliance. We invite you to submit a revised manuscript in line with your proposed Revision Plan.

Thank you for this interesting contribution to Life Science Alliance. We are looking forward to receiving your revised manuscript.

Sincerely,

B. MANUSCRIPT ORGANIZATION AND FORMATTING:

**Revised manuscript LSA-2021-01185
Point-by-point response to reviewers**Reviewer #1 (Evidence, reproducibility and clarity (Required)):

In this MS the authors present wonderful Cryo-ET data, cell biological data and comprehensive proteomic data on the effects of rationally designed aggregating proteins β protein with the aim of to understand the role of the gain-of-function toxicity of aggregates in neurons. The data show that, for these β proteins toxicity relates to a defect in lysosomal function and are in part related to the "sequestration" of the AP-3 complex, a complex involved in internalization from the plasma membrane via the endosomal pathway.

We thank the reviewer for pointing out the strengths of our study, and for the following thorough assessment of the manuscript.

The claim that it is hereby demonstrated that gain-of-function toxicity of protein aggregates is sufficient to recapitulate crucial cellular aspects of protein misfolding diseases, such as impairment of the autophagy-lysosomal pathway and neurotoxicity. First of all, I disagree with this statement as all this paper shows is defects in lysosomal degradation, which is only one of the many cell biological consequences seen in protein misfolding diseases and the typical and highly specific type of lysosomal abnormalities seen with these peptides are in fact only reported for unrelated situations (see specific comments). Moreover, it is questionable from the data whether the lysosomal defects seen here are related to the aggregates of the β proteins or rather due to an unnatural interaction of the soluble β proteins with the proteins in the AP-3 complex.

Based on the reviewer's comment, we have modified the respective sentence about gain-of-function toxicity (Discussion, p. 14, lines 343-344). It now reads: "Our integrated structural, functional and molecular approach demonstrates that gain-of-function toxicity of aggregating proteins can be sufficient to cause lysosomal defects and neurotoxicity."

The type of lysosomal defects described in our study is indeed found in neurodegenerative diseases including synucleinopathies and Alzheimer's disease. In particular, they have been observed in cells from Parkinson's disease patients (Dehay et al., 2012; Usenovic et al., 2012), in the brain tissue of dementia with Lewy bodies (DLB) patients and of α -synuclein transgenic mice (Crews et al., 2010), and in the brain tissue of Alzheimer's disease patients and mouse models (Nixon et al., 2005; Gowrishankar et al., 2015). We have added this information and references on p. 8, lines 162-164.

As stated by the reviewer, our proteomic analyses are focused on the interactors of soluble β proteins, which could range from monomers to soluble oligomers and small aggregates, as pointed out by reviewer #2. We have used two approaches to show that AP-3 μ 1 also interacts with aggregates. First, we demonstrate that AP-3 μ 1 accumulates in the aggregates by colocalization analysis (new data in Fig. 7a-b). Second, we used cellular fractionation and found that AP-3 μ 1 is enriched in the pellet fraction (Fig. 7c-e). Taken together, these data support the notion that AP-3 μ 1 not only interacts with soluble β proteins, but is also sequestered by large aggregates.

In other words: I find the generalizations to other aggregating proteins/disease (even assuming all of this is via "soluble oligomers") unwarranted.

We have made a number of text changes to avoid generalization to other aggregating proteins:

- Results, p. 6: the sentence " β proteins can be used to explore the effects of protein misfolding and aggregation in neuronal cells" is now replaced by " β proteins aggregate and cause toxicity in primary neurons" (lines 116-117).

- P. 7: the statement "The morphology and cellular interactions of β protein aggregates were reminiscent of those formed by natural aggregating proteins" is replaced by "...Despite the distinct characteristics of aggregates formed by different proteins, β protein aggregates reproduced some features of natural amyloids" (lines 148-150).
- Discussion, p. 14: the statement "Gain-of-function toxicity of protein aggregates is sufficient to recapitulate crucial cellular aspects of protein misfolding diseases..." is replaced by "Gain-of-function toxicity of aggregating proteins can be sufficient to cause lysosomal defects and neurotoxicity" (lines 343-344).

So, whereas I find some of the technical aspects of the paper (especially the CryoET data) extremely appealing, it is my opinion that the biological implications/relevance of the data in relation to neurodegenerative protein aggregation diseases is -at most- yet limited.

To strengthen the relevance of our findings for neurodegenerative protein aggregation, we have performed experiments with Tau repeat domain (TauRD) seeding, and analyzed the colocalization of endogenous AP-3 μ 1 with TauRD (new data in Fig. 7f-g). We show that colocalization significantly increases upon seeded aggregation of Tau, suggesting that our findings may also have implications for natural aggregates. In agreement with these data, AP-3 μ 1 also accumulates in the pellet fraction of seeded neurons (new data in Fig. 7h-j). The results are described on p.11-12, lines 274-281.

Furthermore, a number of proteomic studies demonstrated interactions of AP-3 subunits with mutant Huntingtin, and their presence within insoluble mHTT inclusion bodies in mouse models of Huntington's disease (Shirasaki et al., 2012; Hosp et al., 2017). A recent study also identified AP-3 β 2 as one of the most significant interactors of phosphorylated tau in the brain of Alzheimer's disease patients (Drummond et al., 2020). We have expanded the paragraph in the Discussion describing these studies and positioning our findings in light of these publications (p. 15, lines 377-382).

****Specific comments:****

Figure 1: These data nicely show that the artificial peptides can be toxic. Whereas this is important per se, it cannot be taken as a generic statement that "these proteins can be used as a tool to explore the effects of "all" protein misfolding and aggregation in neuronal cells".

We did not claim that β proteins can recapitulate "all" effects of protein misfolding and aggregation. We have rephrased the respective sentence, it now reads: "Altogether, these data indicate that β proteins aggregate and cause toxicity in primary neurons" (Results, p. 6, lines 116-117).

Figure 2: Also here, beautiful images but generic remarks that "We employed cryo-ET to elucidate the mechanism of beta protein toxicity in neurons" and "the morphology and cellular interactions of β protein aggregates were reminiscent of those formed by natural aggregating proteins" (whereas the authors actually mentioned difference between different protein aggregates before this statement) are unwarranted.

We have rephrased the respective sentences as follows:

- Results, p. 6, line 120-121: „We employed cryo-ET to explore the ultrastructure of β protein aggregates and their potentially toxic cellular interactions in neurons“.
- P. 7, lines 148-150: "Thus, despite the distinct characteristics of aggregates formed by different proteins, β protein aggregates reproduced some features of natural amyloids.

Figure 3/4: Again, beautiful images, indeed reminiscent of conditions of impaired lysosomal degradation, but as can be seen in the cited papers, these are found after completely different conditions (chloroquine or imipamide hydrochline poisoning; mitochondrial chain defects) and as such these very typical lysosomal structure have -to my knowledge- never been reported

for as very abundant features of neurodegenerative diseases associated with intracellular protein aggregation.

This type of lysosomal defects has been observed e.g. in Alzheimer's disease and Parkinson's disease patients and mouse models (Nixon et al., 2005; Gowrishankar et al., 2015; Dehay et al., 2012; Usenovic et al., 2012; Crews et al., 2010). This information and references are now mentioned on p. 8, lines 162-164.

Figure 5 (minor): Panel A is of insufficient quality to determine whether or not the β protein aggregates did or did not affect autophagic flux. If anything, the data may further suggest basal flux to be slightly higher in cells with β protein aggregates. Is this significant and if so, what does this imply. Also, I am not sure the authors can conclude for the rest of the data in this figure that the data but confirm a "block in the autophagy flux due to decreased lysosomal acidity". I think a more neutral conclusion is that "intra-lysosomal function/degradation seems affected".

We thank the reviewer for pointing out these unclarities. We have replaced the image in Fig. 5a. The difference in LC3 levels between mCherry and β 23-mCherry in Fig. 5b is not significant. This is now indicated on the graph.

As suggested by the reviewer, we have rephrased the respective conclusion, it now reads (p. 9, lines 201-202): "These results ... confirm a defect in lysosomal degradation".

Figure 6/7 (major):

First, it must be stated that immunoprecipitation experiments are notoriously dangerous when using aggregation-prone proteins as a bait. From the methods sections it is difficult to assess whether or not aggregated β protein is included in the lysates used for IP or discarded? If included, can one exclude post-lysis artifacts? If excluded, are we not missing essential (other) interactors?

Immunoprecipitation experiments have been successfully used in many studies of aggregation-prone proteins such as mutant Huntingtin and C9ORF72 dipeptide repeats (e.g., Shirasaki et al, Neuron 2012, PMID 22794259; Sivadasan et al., Nat Neurosci 2016, PMID 27723745; May et al. Acta Neuropathol 2014, PMID 25120191; also see review by Laskowska et al., J Proteomics 2019, PMID 30529741). The relevance of the interactors identified in these studies was demonstrated in extensive follow-up experiments in animal models and human tissue, yielding ample mechanistic insights into pathological mechanisms of aggregation-prone proteins. We therefore believe that this approach, despite the limitations mentioned by the reviewer, is suitable for investigating disease-relevant pathways, in particular if the findings are validated with independent strategies such as cell biological assays and cryo-ET. In addition, we performed affinity enrichment using relatively mild conditions (5% Glycerol, 150mM NaCl, 50 mM Tris HCl, with a low concentrated detergent of 0.1-0.25% NP-40, 4°C assay temperature) that have previously been shown to retain physiologically relevant protein-protein interactions. We discussed the utility of this mild affinity enrichment strategy in contrast to more harsh affinity purification strategies previously (Keilhauer et al., Mol Cell Proteomics 2015, PMID 25363814). While affinity enrichment is better suited to retain physiologically relevant interactions, it requires very precise quantification of peptide and protein differences to quantify small fold changes between samples sets. This is enabled by state-of-the-art LC-MS/MS methods in conjunction with advanced normalization approaches such as Max LFQ (Cox et al., Mol Cell Proteomics 2014; PMID 24942700).

The IPs were performed with soluble fractions. This is now mentioned on p. 11, lines 266-267: "...the interactome experiments were performed with the soluble fraction of neuronal lysates..." As pointed out by reviewer #2, this fraction might include oligomers and small aggregates. It is true that additional interactors specific to larger insoluble aggregates

would probably be missed in this screen, and might be an interesting direction for further studies.

As I infer from the authors' statement that these experiments were done "at a time point when β proteins are still largely soluble and do not cause massive cell death", I assume we are looking here at a situation where aggregates are indeed to be excluded. This thus implies the identified interactome is entirely with "soluble" β peptides. If correct, this even more so asks for a very careful "generic translation" of the resulting lysosomal defects as a general aspect of protein aggregation diseases. In other words, the effects are totally unrelated to the aggregation phenomenon per se, but a pure result of a rather (un)fortunate interaction of an artificial peptide with a number of intracellular proteins, including AP-3 μ 1, that next leads to abnormal trafficking of transmembrane proteins and hence abnormal lysosomes.

We would like to point out that small aggregated species such as soluble oligomers have been shown to drive toxicity in neurons and thus the interaction of oligomers can be a very relevant part of the pathological cascade. This is supported by the consistent evidence of lysosomal impairments by β proteins across the different assays presented in our paper.

To evaluate the relevance of the AP-3 μ 1 / β protein interaction for larger aggregates, we have performed colocalization analysis of endogenous AP-3 μ 1 and β protein aggregates in transfected neurons at DIV 10+2 (analyses at later time points were precluded by β protein toxicity). We observe a clear colocalization between the two proteins (Fig. 7a-b). This is corroborated by cell fractionation experiments showing that AP-3 μ 1 is enriched in the insoluble fraction at DIV 10+2, when significant colocalization with the aggregates was observed by immunostaining (Fig. 7e). Taken together, these results indicate that AP-3 μ 1 not only engages in an aberrant interaction with the soluble form of β proteins, but is also sequestered by large β protein aggregates.

However, only a few lines further, the heading of the next paragraph states "AP-3 μ 1 is sequestered by β protein aggregates, compromising the integrity of the AP-3 complex". For this, figure 6i-j are suggested to show colocalization of AP-3 μ 1 with the β protein aggregates (puncta), but these images are not very convincing.

We provide new images in panel 6i (now panel 7a), which show a clearer colocalization of AP-3 μ 1 with β protein aggregates.

Figure 7 next is suggested to demonstrate that AP-3 μ 1 was reduced in the soluble fraction of β 4-mCherry neurons by 30% (Fig. 7a-c), but this is not convincing at all as also total levels seem lower under these conditions. Also, pellet fractions should be shown to demonstrate this truly is the case. In addition, it is assumed that such a 30% decrease by this presumed sequestration would suffice to cause a partial loss of function, since AP-3 μ 1 is not highly abundant in neurons. Whilst such an argumentation seems to be not valid per se anyhow, it also should be demonstrated (e.g. by siRNA approaches) that just a 30% reduction (and not only a near to complete reduction as in Mocha cells) in levels is sufficient to explain the lysosomal defects seen here. Yet, the rescue of the toxicity by overexpressing AP-3 μ 1 (figure S8e) does support the conclusion that at least in the case of these artificial peptide, the sequestration of AP-3 μ 1 and associated lysosomal defects mediate, at least in part, the toxicity of β proteins (but not aggregates) in neurons.

We have included the pellet fraction, it is now shown in Fig. 7c, e. As we observed accumulation of tubulin in the pellet in the presence of β 4-mCherry, the levels of AP-3 subunits in the soluble and pellet fractions are now normalized to the respective subunits in the whole lysate instead of normalizing to tubulin. These results reveal a significant accumulation of AP-3 μ 1 in the pellet. Although AP-3 μ 1 levels in the soluble fraction are not significantly reduced, it should be kept in mind that the AP-3 μ 1 found in the soluble

pool is partially non-functional due to its aberrant interaction with the soluble aggregating species, as revealed by our mass spectrometry data.

To further assess the consequences of AP-3 μ 1 reduction for the assembly and stability of the AP-3 complex as a whole, we have performed Western blots for the other AP-3 subunits, and observed a pronounced reduction in both AP-3 δ 1 and AP-3 σ 1 (Fig. 8a-b), which occurs in parallel aggregate formation. These data suggest that the integrity of the AP-3 complex is compromised. In addition, we have attempted performing IPs for single AP-3 subunits and quantifying the levels of the other co-precipitating subunits to estimate how much of the intact heterotetrameric complex is present in the cells (as in Peden et al., JCB 2002). However, due to technical difficulties with the IPs we did not manage to obtain conclusive results within the time frame of the revisions.

We agree with the reviewer that a partial knockdown of AP-3 μ 1 would be the desired setting for testing our hypothesis. Unfortunately, our extensive attempts to perform knockdown of AP-3 μ 1 in neurons were unsuccessful. We therefore resorted to mocha cells as an alternative AP-3 loss-of-function model. Considering that the reduction in AP-3 δ 1 in β protein neurons was as large as 65%, the full knockout Mocha cells are still a relevant model.

Reviewer #1 (Significance (Required)):

Whereas I find the major part of the data technically of high quality and the conclusion correct that part of the toxicity of these artificial peptides may be indeed do to sequestration of AP-3 μ 1 and a related protein trafficking to lysosomal organelles, I think that the significance of this for the mechanisms of the many different protein aggregation disease could be limited. Even if all is related to the aggregation of these peptides (which to me is not clear from the current data) it remains to be seen (and certainly is not clear) if other aggregates would do the same. In fact, in my view this is even highly unlikely and therefore the generalisation of these data to protein aggregation disease in general (as it is claimed) is (yet) unwarranted.

We now show that endogenous AP-3 μ 1 also colocalizes with TauRD aggregates in neurons (Fig. 7f-g), and that it accumulates in the insoluble fraction upon seeding of TauRD aggregates (Fig. 7h-j). These new data suggest that our findings can be extended to natural aggregates. We have also included references on the sequestration of AP-3 subunits and lysosomal defects caused by natural amyloids (Discussion, p.15, lines 377-385).

We have rephrased the claims about generalization of the data (see above).

****Referee Cross-commenting****

Whereas all reviewers (including I myself) seem to be equally impressed by the superb cryoEM tomography data, we should not be distracted by these beautiful images as if they are the first to demonstrate gain-of-function effects resulting from protein aggregation.

I respectfully disagree with reviewer 2 on this statement as also expression of many other aggregating proteins (even some more closely related to disease than the artificial peptides use here) have already (and even more clearly than this paper) demonstrated the toxic gain-of-function that aggregates CAN exert (whether this causes disease may still be debated, but this would be a discussion that goes beyond what the current paper addresses). Even more so, however, in this paper it seems doubtful whether the toxicity is truly related to the aggregates formed by the peptides or that toxicity is caused by soluble peptides bind to AP-3 μ 1 and hereby have some dominant-negative effects on the complex that protein must form to exert its function. Rescue of toxicity by co-expressing AP-3 μ 1 together with the peptide is

also consistent with such a scenario. Experiments with a non-aggregating peptide that does bind to AP-3 μ 1 could be used to discriminate this, but it may be hard to generate such.

The observation that the onset of neuronal cell death correlates with the appearance of aggregates (Fig. S1a-e) supports the role of aggregates in neurotoxicity. As pointed out by the reviewer, we cannot exclude the possibility that non-aggregated soluble β proteins might contribute to toxicity as well. However, it should be noted that also the soluble fraction contains aggregating species such as soluble oligomers. The potential role of the soluble species is acknowledged in the Discussion (p. 14, lines 339-341): “Apart from fibrillar aggregates, soluble oligomers may also play a role in the effects observed here, as a large fraction of the β proteins remained soluble at the time point of the analyses (Supplementary Fig. S1).”

I also respectfully disagree with referee #3 that this study makes a significant advance in integrating structural and mechanistic studies to understand how protein aggregates cause cellular toxicity. Rather, I think that a unified theory for all different protein aggregation diseases (assuming they are driven by aggregation) and a linear (single) mode of action of these aggregates (let alone a single protein as target) to explain their pathogenesis is to be strongly doubted given the existing literature. Each aggregate arising from different disease-associated protein is likely to sequester different sets of proteins and even for the same disease-associated protein this may differ from where the aggregates start to form. In that context, I also like to emphasise that the type of structures found in this study have not been reported as typical for most aggregation diseases (although I must admit that people may not have looked careful enough with similar advanced technology). Yet, the cited studies that do seem to have reported such structures are NOT for protein aggregation diseases but related to toxic or mitochondrial diseases. So, for this aspect, I support the comment of reviewer 2 that the impact and significance of these findings would be much improved if the authors tested natural neuropathological amyloid, but I doubt whether or not this will be a reasonable request (also given my scepticism that it will actually not work).

The type of structures described in our study is indeed found in neurodegenerative diseases such as Alzheimer’s disease and synucleinopathies, as explained above.

We agree with the reviewer that a single explanation for the toxicity of different types of aggregates is unlikely, and that each of them has a distinct set of interactors. It is also true that the toxicity of any type of aggregate is multifactorial and involves a variety of different pathways, and even the same aggregating protein might exist in different strains with distinct cellular effects. In this study, we suggest that sequestration of AP-3 μ 1 might be one of the factors contributing to toxicity, and do not claim that it is the only important factor. The discussion already contains the following disclaimers in this regard:

- “The toxicity mechanisms triggered by aggregating proteins are multifactorial and involve various cellular processes” (p. 14, lines 348-349).
- „It is also possible that impairment of cellular processes beyond lysosomal trafficking contributes to the deleterious effects of β proteins” (p.15, lines 374-375).

As recommended by reviewer #2 and now supported by reviewer #1, we have investigated recruitment of AP-3 μ 1 to aggregates of TauRD, a natural amyloidogenic protein. We show that AP-3 μ 1 is also present within these aggregates (Fig. 7f-g), and accumulates in the pellet fraction in cell fractionation experiments (Fig. 7h-j). These data strengthen the relevance of our findings for neurodegenerative protein aggregation.

Finally, I would like to stress that I do not think the artificial peptides used are irrelevant to study protein aggregation. In fact, they already have and could provide more insights in how cells handle aggregation-prone proteins (even though also this may differ depending on the nature of the protein). However, their significance for deriving a generalisable mechanism for aggregate toxicity (as this study claims), I think they have less value.

As outlined above, we have rephrased the statements generalizing our findings to other types of aggregates.

Reviewer #2 (Evidence, reproducibility and clarity (Required)):

In this manuscript, Dr. Dudanova and colleagues aim at identifying the mechanistic link between amyloid-like protein aggregation and lysosomal dysfunction, which is frequently observed in various neurodegenerative diseases. As it is difficult to disentangle loss-of-function and gain-of-function effects resulting from protein aggregation, the authors used artificial constructs allowing the expression of non-native beta-aggregating polypeptides with no other cellular function. Hence, only the toxic gain-of-function effects, which are likely to be common to different amyloid aggregates in different pathologies, are investigated here. Using an impressive combination of techniques (cell culture, immunofluorescence and live cell microscopy, cryo-ET, biochemistry, mass-spec) and different models (primary neurons, HeLa cells), they convincingly show that beta-proteins aggregation impairs proper lysosomal function and results in neurotoxicity. The authors identify AP-3 μ 1, a component of the AP-3 adaptor complex, as a main target of beta-protein aggregates.

The manuscript is well written. The experiments are clearly described and well conducted and the data mostly support the conclusions made by the authors, pending clarifications (see major comments below). In my opinion, the major weakness of this paper is that it is unclear to which extent the findings described here are generalizable (see significance paragraph).

We thank the reviewer for appreciating our approach and for the supportive comments.

****Major comments:****

- The use of different cell types, transfected/transduced at DIV10 or DIV6 and recovered 1 to 6 days after transfection/transduction is confusing at times. For some experiments (e.g. Fig S3), it is unclear at which stage the cells were taken. For some apparently identical conditions (transfected cortical neurons DIV6+1), the beta-protein form aggregates (e.g. Fig 2) or appear to remain mostly soluble (e.g. Fig 4). I suggest the authors clarify these issues and maybe summarize the different cell types/constructs/time points/aggregation behavior in a Table S2.

We thank the reviewer for pointing out these unclarities. We have specified the time point in the legend to Fig. S3.

The different appearance of aggregates in Fig. 4a compared to e.g. Fig. 1 is at least partially due to technical issues: while the images in Fig. 1 were acquired from fixed cells with a confocal microscope, the ones in Fig. 4 are from live cells acquired with a widefield microscope, which has a lower resolution and higher out-of-focus noise. Nevertheless, aggregates were typically observed in both assays. We have replaced the images in Fig. 4a to illustrate them more clearly.

As suggested, we have summarized the experimental conditions in a new Supplementary Table S1 (p. 43).

- The interactors of beta-proteins were identified in DIV10+4 primary cortical neurons, a time point where the proteins are mostly soluble (Fig S1). Hence, it is unclear whether these interactors are "present in the aggregates" (as stated by the authors in page 9) and whether these results can be linked to the beta-aggregates inclusions analyzed by cryoET. It is noteworthy that the contribution of AP-3 μ 1 to beta-proteins-interacting complexes decreases as the aggregation propensity of the beta-proteins increases (beta4: 26% - beta17: 17% - beta23: 10%). Would it be possible to demonstrate the sequestration of AP-3 μ 1 (or other major

interactors) in beta-aggregates (as suggested by fig. 7) within cells using immunogold staining and EM? The differential centrifugation (4,000-14,000 x g) used to separate 'soluble' and 'insoluble' fractions is not sufficiently informative as oligomers or small aggregates will likely remain in the 'soluble' fraction and significantly contribute to the effects and interactions described by the authors.

We have rephrased the sentence (now on p. 11, lines 254-255) as follows: "This indicates that their presence in the interactome..." Similarly, we have rephrased the next sentence, which now reads "their enrichment in the interactome" instead of "in the aggregates". It is indeed possible that some of the observed effects might be due to soluble β protein species, as acknowledged in the Discussion on p. 14, lines 339-341: "Apart from fibrillar aggregates, soluble oligomers may also play a role in the effects observed here, as a large fraction of the β proteins remained soluble at the time point of the analyses".

While the contribution of AP-3 μ 1 to β protein-interacting complexes decreases with increasing aggregation propensity of the β proteins, as correctly pointed out by the reviewer, note that this contribution correlates well with the toxicity of the β proteins. Unlike in HEK cells, we find that β 4 in neurons is more toxic than β 23 (Fig. 1d and S1e).

EM experiments with immunogold labeling is an interesting suggestion, but we were unfortunately not able to perform such studies in the time scope of the revision.

To demonstrate that AP-3 μ 1 is also sequestered by insoluble aggregates, we have included Western blots of the pellets, which show accumulation of AP-3 μ 1 in the insoluble fraction in β protein cells (Fig. 7 c, e). Moreover, we have analyzed β protein neurons by immunostaining, and observed a clear colocalization of endogenous AP-3 μ 1 with β protein aggregates (new data in Fig. 7a-b).

- AP3 μ 1 is a very interesting hit and I understand why the authors focused on this interactor to provide a mechanistic link between protein aggregation and lysosomal dysfunction. Nevertheless, I found it frustrating that the authors did not at least discuss the other major hits they identified, particularly the ones that appear to contribute significantly to beta-protein-interacting complexes (e.g. ccdc88a; Fig 6e-h).

Following the reviewer's suggestion, we have added a description of two other major interactors on p. 10, lines 236-240: "Other prominent common interactors were Ccdc88a and Crmp1, two proteins involved in neuronal development and synaptic plasticity (Enomoto et al., 2009; Nakai et al., 2014; Yamashita and Goshima 2012). Of note, Ccdc88a plays a role in intracellular trafficking, among other functions (Le-Niculescu et al., 2005). Crmp1 is also found within mutant Huntingtin inclusions, and suppresses neurotoxicity in Huntington's disease models (Stroedicke et al., 2015)." Many of the identified proteins are indeed interesting targets for future studies, but following up on all of them is outside the scope of this manuscript.

****Minor comments:****

- Table S1 is unclear. For instance, I do not understand how Table S1 shows that "beta protein aggregates displayed fibrillar morphology" (page 6).

The table (now called Supplementary Table S2) shows numbers of independent experiments, cells and analyzed subcellular structures for all cryo-ET experiments. We have changed the title of the table to "Size of experimental samples for cryo-ET experiments". We have rephrased the mentioned sentence (now on p. 7, lines 129-130) as follows: "...All β protein aggregates displayed fibrillar morphology (for numbers of analyzed aggregates, see Supplementary Table S2)". We have also rephrased accordingly the sentences on p. 8 and 9 containing references to this table.

- Page 7: *The effect of chloroquine on lysosomal degradation may not be due to increased lysosomal pH (e.g. Mauthe et al. autophagy 2018).*

We have deleted the respective part of the sentence and included the reference Mauthe et al. 2018. The sentence now reads (p. 8, lines 181-184): "First, we quantified LC3-II turnover in β 23-mCherry transfected HeLa cells (Fig. S5a) in the presence and absence of 50 μ M chloroquine, which blocks lysosomal degradation (Mauthe et al., 2018; Poole and Ohkuma, 1981).

- Fig S1b: *the amount of proteins in the pellet fraction is not shown below the plots*

We have included a graph showing quantification of β proteins in the pellet (Fig. S1d).

- Fig S7f: *I suggest showing an example of a known abundant protein.*

Thank you for the suggestion, we have indicated Hsp90 on the graph as an example of a known abundant protein.

Reviewer #2 (Significance (Required)):

This paper provides a novel mechanistic link between the aggregation of a beta-rich protein and lysosomal dysfunction, which is frequently observed in several age-related neurodegenerative diseases. This is a very nice paper that will appeal to a wide range of scientists interested in neurodegeneration, protein misfolding and degradation, autophagy and lysosomal dysfunctions. The "integrated structural, functional and molecular approach" (page 12) is indeed inspiring.

However, as the authors note in the discussion section, the artificial beta-proteins used in this study differ from natural amyloids, for instance by the lack of globular/functional domains that may modulate the aggregation propensity and toxicity of the beta-rich domains. This is exemplified by the fact that at first glance, the organization of the beta-protein inclusions analyzed by cryoET appear to be somehow different from that of the alpha-synuclein inclusions previously reported by the authors (Trinka et al. 2021). This point is not discussed in the paper.

We agree with the reviewer that β protein aggregates are in some ways different from alpha-synuclein inclusions. Indeed, all types of aggregates display some unique features (Bauerlein et al., Cell 2017; Gruber et al., PNAS 2018; Guo et al., Cell 2018; Trinka et al., Nat Comm 2021), and even the same aggregating protein can form aggregates of different structure (e.g., Shi et al., Nature 2021). To acknowledge this, we have rephrased the conclusion on p. 7 (lines 148-150), which now reads: "Thus, despite the distinct characteristics of aggregates formed by different proteins, β protein aggregates reproduced some features of natural amyloids".

In my opinion, the impact and significance of these findings would be much improved if the authors tested if at least one natural neuropathological amyloid (e.g. alpha-synuclein, Tau) sequesters AP3 components (or other interactors identified here) and similarly cause lysosomal dysfunction. To avoid the loss-of-function problem and still have a "natural" protein, the authors could also try expressing a heterologous natural amyloid (e.g. a yeast prion) and compare its behaviour to that of their beta-proteins.

As suggested by the reviewer, we have tested whether TauRD also sequesters AP-3 subunits. We observe significant colocalization of AP-3 μ 1 with TauRD when aggregation is induced by adding recombinant TauRD seeds (new data in Fig. 7f-g). These results are

also corroborated by cell fractionation experiments, which reveal accumulation of AP-3 μ 1 in the insoluble fraction of seeded neurons (new data in Fig. 7h-j). The results are described on p.11-12, lines 274-281.

Moreover, the significance of our findings is supported by proteomic studies of natural aggregating proteins. Thus, AP-3 β 2 was found to be an interactor of soluble Huntingtin (Shirasaki et al., Neuron 2012), and together with AP-3 δ 1 it was also identified as a component of insoluble mutant Huntingtin inclusions in the mouse brain (Hosp et al., 2017). A recent proteomic study identified AP-3 β 2 as one of the most significant interactors of phosphorylated tau in the brain of Alzheimer's disease patients (Drummond et al., 2020). We have expanded the paragraph in the Discussion describing these studies (p. 15, lines 377-382).

Reviewer's expertise: protein misfolding and quality control, molecular chaperones, prions and amyloids, neurodegenerative diseases, biochemistry.

****Referee Cross-commenting****

I thank reviewer #1 for his thoughtful comments. I apologize for maybe not being clear enough, but I did not mean to state that this study is the first to demonstrate gain-of-function exerted by protein aggregates. I believe reviewer #1 and myself agree on most points, specifically as to whether the identified interactors bind to aggregated or soluble species (and which soluble species), and to which extent the findings are generalizable to disease-related amyloids. I agree with reviewer #1 that different protein aggregates may bind different sets of proteins and cause toxicity through different mechanisms.

Reviewer #3 (Evidence, reproducibility and clarity (Required)):

****Summary:****

Riera-Tur et al demonstrate that synthetic amyloid-forming proteins (beta proteins) cause toxicity in primary neurons. They then use cryo-ET, proteomics, and cell biology to demonstrate that beta protein aggregates impair lysosomal morphology and function. They show that beta proteins sequester subunits of the AP complex within aggregates and thereby impair lysosomal function and cause cell death.

The data is generally convincing, subject to the following revisions:

We thank the reviewer for the positive comments.

(1) In Figure 1c (and at various other points throughout the manuscript) the authors either show only examples of beta23 (or beta4) cells. It would be more convincing to see representative images of all conditions (mCherry, beta4, and beta23). Similarly in Fig. 1e, it would be helpful to see an example of beta4's effect on neuronal morphology.

Following the reviewer's suggestion, we have added images of the respective other protein wherever possible throughout the manuscript. As we observed overall very similar effects of β proteins, some assays were performed with only one of the two proteins, and in some cases such as in Fig. 1c we only acquired high-resolution images for one of the β protein conditions.

- Fig. 1e: β 4-mCherry added
- Fig. 2: β 23-mCherry moved from the supplements to the main figure and is now shown side-by-side with β 4-mCherry
- Fig. 4a: β 23-mCherry added
- Fig. 8c (previously 7d): β 4-mCherry added

- Fig. S4: β 23-mCherry added (panels g-i)

(2) In Figure 2 and the accompanying discussion of the results, the authors presently only mention the average width of fibrils. It would be helpful to (1) show tomogram of beta23-expressing cells, and (2) to show quantitation of fibrils diameters in Figure 2, eg a histogram of fibril diameters. It may also be useful to quantitate average branch length, although I do not think this is essential.

We thank the reviewer for the suggestions. We have moved the tomogram of a β 23-mCherry neuron from Supplementary Fig. S3 to Fig. 2 (panels c-d). We have also added a histogram of fibril diameters (panel e).

Quantifying average branch length is an interesting idea, but is very difficult to perform for two reasons. First, there is a missing wedge of information in the tomograms of approximately 60° , and structures that are perpendicular to the electron beam cannot be resolved. Second, the fibrils frequently change direction. As they cannot be tracked through the missing wedge, they are randomly cut short, precluding us from reliably calculating their average length.

(3) On pg. 6, the authors state "the fibrillar network encapsulated additional electron-dense structures that may correspond to cellular proteins sequestered by the aggregates, such as ribosomes." However, the authors include no data to support this claim at this point in the manuscript, although they eventually do. I would suggest to eliminate this sentence, or if there is prior precedent for this observation (as suggested by including citations in the sentence), make clear that the statement at this point is based on prior literature rather than their observations.

We apologize if this point was not clear. The ribosomes are visible on the tomogram in Fig. 2a, and are labeled in yellow on the corresponding 3D rendering in Fig. 2b. We now refer to Fig. 2b in the text on p. 7, line 136, and hope this is sufficiently clear.

(4) On pg. 6., the authors state that 'beta protein fibrils did not appear to deform cellular membranes.' While I agree with this statement, it would be better to include a 'no beta protein' control to compare in this panel. Perhaps it would be good to combine Figs. 2 and 3, as Fig. 3a-b shows morphology of cells not expressing a beta protein?

We thank the reviewer for the suggestion. We have included an inset in Fig. 2c (lower inset) showing intracellular membranes that are not in direct contact with the fibrils as a control. We prefer not to combine Figures 2 and 3, because they have different messages (ultrastructure of the aggregates vs. lysosomal defects).

Reviewer #3 (Significance (Required)):

The study makes a significant advance in integrating structural and mechanistic studies to understand how protein aggregates cause cellular toxicity.

Referee Cross-commenting

Thanks for your explanation. I indeed noted we agree on most points. To me this implies, however, that -albeit a beautifully illustrated and microscope technically superb paper- neither the title (Amyloid-like aggregates cause lysosomal defects in neurons via gain-of- function toxicity) nor the main generic conclusion (Altogether, our results highlight the link between protein aggregation and neurotoxicity, pointing to lysosomes as particularly vulnerable organelles) is supported by the data.

I find it important to state this as these (and many other) generalisations on aggregation

diseases and via which (authors' favourite) organelles they lead to pathology have confused the field more than that they have helped it forward.

We have rephrased the title: "Amyloid-like aggregating proteins cause lysosomal defects in neurons via gain-of-function toxicity".

We have rephrased the conclusion in the abstract, it now reads: "Altogether, our results highlight the link between protein aggregation, lysosomal impairments and neurotoxicity" (lines 50-51).

In addition, we have made a number of text changes following recommendations of Reviewer #1 to avoid generalizations to other aggregating proteins:

- Results, p. 6, lines 116-117: the sentence " β proteins can be used to explore the effects of protein misfolding and aggregation in neuronal cells" is now replaced by " β proteins aggregate and cause toxicity in primary neurons".
- P. 7, lines 148-150: the statement "The morphology and cellular interactions of β protein aggregates were reminiscent of those formed by natural aggregating proteins" is replaced by "...Despite the distinct characteristics of aggregates formed by different proteins, β protein aggregates reproduced some features of natural amyloids".
- Discussion, p. 14, lines 343-344: the statement "Gain-of-function toxicity of protein aggregates is sufficient to recapitulate crucial cellular aspects of protein misfolding diseases..." is replaced by "Gain-of-function toxicity of protein aggregates can be sufficient to cause lysosomal defects and neurotoxicity".

In addition to the changes requested by the reviewers, we have added individual data points to the bar graphs throughout the figures in the revised version of the paper.

November 29, 2021

RE: Life Science Alliance Manuscript #LSA-2021-01185R

Dr. Irina Dudanova
Max Planck Institute of Neurobiology
Am Klopferspitz 18
Martinsried 82152
Germany

Dear Dr. Dudanova,

Thank you for submitting your revised manuscript entitled "Amyloid-like aggregating proteins cause lysosomal defects in neurons via gain-of-function toxicity". We would be happy to publish your paper in Life Science Alliance pending final revisions necessary to meet our formatting guidelines.

- please add ORCID ID for secondary corresponding author-they should have received instructions on how to do so
- please use capital letters when introducing panels in figures, their legends, and callouts in the manuscript text
- we encourage you to revise the figure legends for figures 2, 3 such that the figure panels are introduced in alphabetical order
- Please indicate molecular weight next to each protein blot
- please add callouts for Figures S2A-F and S4G and I to your main manuscript text
- please provide a statement indicating approval (and by what authority) to perform experiments using CD-1 and C57BL/6 embryos.

A. FINAL FILES:

B. MANUSCRIPT ORGANIZATION AND FORMATTING:

Sincerely,

Reviewer #2 (Comments to the Authors (Required)):

I would like to thank the authors for taking into account and addressing the reviewers comments, including my own. I found the revised manuscript to be very much improved and strengthened by the new data. In my opinion, the manuscript is now acceptable for publication in Life Science Alliance.

December 3, 2021

RE: Life Science Alliance Manuscript #LSA-2021-01185RR

Dr. Irina Dudanova
Max Planck Institute of Neurobiology
Am Klopferspitz 18
Martinsried 82152
Germany

Dear Dr. Dudanova,

Thank you for submitting your Research Article entitled "Amyloid-like aggregating proteins cause lysosomal defects in neurons via gain-of-function toxicity". It is a pleasure to let you know that your manuscript is now accepted for publication in Life Science Alliance. Congratulations on this interesting work.

DISTRIBUTION OF MATERIALS:

Again, congratulations on a very nice paper. I hope you found the review process to be constructive and are pleased with how the manuscript was handled editorially. We look forward to future exciting submissions from your lab.

Sincerely,
